# Causal Imitability Under Context-Specific Independence Relations

**Fateme Jamshidi**
EPFL, Switzerland
fateme.jamshidi@epfl.ch

**Sina Akbari**
EPFL, Switzerland
sina.akbari@epfl.ch

**Negar Kiyavash**
EPFL, Switzerland
negar.kiyavash@epfl.ch

## Abstract

Drawbacks of ignoring the causal mechanisms when performing imitation learning have recently been acknowledged. Several approaches both to assess the feasibility of imitation and to circumvent causal confounding and causal misspecifications have been proposed in the literature. However, the potential benefits of the incorporation of additional information about the underlying causal structure are left unexplored. An example of such overlooked information is context-specific independence (CSI), i.e., independence that holds only in certain contexts. We consider the problem of causal imitation learning when CSI relations are known. We prove that the decision problem pertaining to the feasibility of imitation in this setting is NP-hard. Further, we provide a necessary graphical criterion for imitation learning under CSI and show that under a structural assumption, this criterion is also sufficient. Finally, we propose a sound algorithmic approach for causal imitation learning which takes both CSI relations and data into account.

## 1 Introduction

Imitation learning has been shown to significantly improve performance in learning complex tasks in a variety of applications, such as autonomous driving [27], electronic games [17, 41], and navigation [34]. Moreover, imitation learning allows for learning merely from observing expert demonstrations, therefore circumventing the need for designing reward functions or interactions with the environment. Instead, imitation learning works through identifying a policy that mimics the demonstrator's behavior which is assumed to be generated by an expert with near-optimal performance in terms of the reward. Imitation learning techniques are of two main flavors: *behavioral cloning* (BC) [42, 30, 24, 25], and *inverse reinforcement learning*[1] (IRL) [26, 1, 37, 44]. BC approaches often require extensive data to succeed. IRL methods have proved more successful in practice, albeit at the cost of an extremely high computational load. The celebrated generative adversarial imitation learning (GAIL) framework and its variants bypass the IRL step by occupancy measure matching to learn an optimal policy [15].

Despite recent achievements, still in practice applying imitation learning techniques can result in learning policies that are markedly different from that of the expert [9, 7, 22]. This phenomenon is for the most part the result of a distributional shift between the demonstrator and imitator environments [11, 31, 12]. All aforementioned imitation learning approaches rely on the assumption that the imitator has access to observations that match those of the expert. An assumption clearly violated when unobserved confounding effects are present, e.g., because the imitator has access only to partial observations of the system. For instance, consider the task of training an imitator to drive a car, the causal diagram of which is depicted in Figure 1a. $X$ and $Y$ in this graph represent the action taken by the driver and the latent reward, respectively. The expert driver controls her speed ($X$) based on the speed limit on the highway (denoted by $S$), along with other covariates such as weather conditions,

---

[1]Also referred to as inverse optimal control in the literature.

37th Conference on Neural Information Processing Systems (NeurIPS 2023).

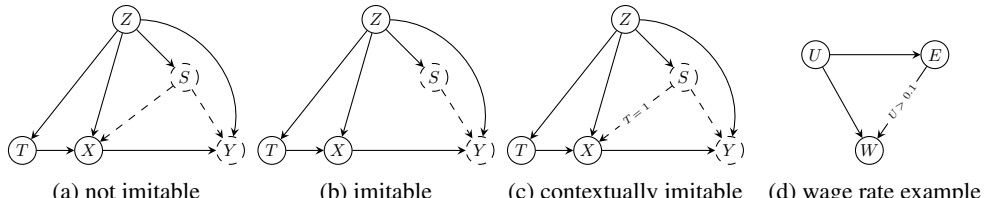

Figure 1: Example of learning to drive a car. (a) Imitation is impossible, as the imitator has no access to the speed limit. (b) The car has automatic cruise control, which makes the expert actions independent of the speed limit. (c) Driver actions are independent of speed limit only in the context of heavy traffic. (d) A simple example of a CSI relation in economics.

brake indicator of the car in front, traffic load, etc. We use two such covariates in our example: $Z$ and $T$. An optimal imitator should mimic the expert by taking actions according to the expert policy $P(X|S, Z, T)$. However, if the collected demonstration data does not include the speed limit ($S$), the imitator would tend to learn a policy that averages the expert speed, taking only the other covariates ($Z$ and $T$) into account. Such an imitator could end up crashing on serpentine roads or causing traffic jams on highways. As shown in Figure 1a, the speed limit acts as a latent confounder[2].

One could argue that providing more complete context data to the imitator, e.g. including the speed limit in the driving example, would resolve such issues. While this solution is not applicable in most scenarios, the fundamental problem in imitation learning is not limited to causal confounding. As demonstrated by [12] and [43], having access to more data not only does not always result in improvement but also can contribute to further deterioration in performance. In other words, important issues can stem not merely from a lack of observations but from ignoring the underlying causal mechanisms. The drawback of utilizing imitation learning without considering the causal structure has been recently acknowledged, highlighting terms including causal confusion [12], sensor-shift [13], imitation learning with unobserved confounders [43], and temporally correlated noise [36].

Incorporating causal structure when designing imitation learning approaches has been studied in recent work. For instance, [12] proposed performing targeted interventions to avoid causal misspecification. [43] characterized a necessary and sufficient graphical criterion to decide the feasibility of an imitation task (aka *imitability*). It also proposed a practical causal imitation framework that allowed for imitation in specific instances even when the general graphical criterion (the existence of a $\pi$-backdoor admissible set) did not necessarily hold. [36] proposed a method based on instrumental variable regression to circumvent the causal confounding when a valid instrument was available.

Despite recent progress, the potential benefits of further information pertaining to the causal mechanisms remain unexplored. An important type of such information is *context-specific independence* (CSI) relations, which are generalizations of conditional independence. Take, for instance, the labor market example shown in Figure 1d, where $W$, $E$, and $U$ represent the wage rate, education level, and unemployment rate, respectively. The wage rate $W$ is a function of $E$ and $U$. However, when unemployment is greater than $10\%$, the education level does not have a significant effect on the wage rate, as there is more demand for the job than the openings. This is to say, $W$ is independent of $E$ given $U$, only when $U > 10\%$. This independence is shown by a label $U > 0.1$ on the edge $E \rightarrow W$. Analogously, in our driving example, the imitator would still match the expert's policy if there was heavy traffic on the route. This is because, in the context that there is heavy traffic, the policy of the expert would be independent of the speed limit. This context-specific independence between the speed limit $S$ and the action $X$ given $T = 1$ (heavy traffic) is indicated by a label ($T = 1$) on the edge $S \rightarrow X$ in Figure 1c. This label indicates that the edge $S \rightarrow X$ is absent when $T = 1$. The variables $T$ in Figure 1c and $U$ in Figure 1d are called *context variables*, i.e., variables which induce conditional independence relations in the system based on their realizations.

CSIs can be incorporated through a refined presentation of Bayesian networks to increase the power of inference algorithms [8]. The incorporation of CSI also has ramifications on causal inference. For instance, [39] showed that when accounting for CSIs, Pearl's do-calculus is no longer complete for causal identification. They proposed a sound algorithm for identification under CSIs. It is noteworthy

---

[2]On the other hand, if the car had automatic cruise control, the expert policy would be independent of the speed limit, resolving imitation issues (refer to Figure 1b.)

that unlike conditional Independence (CI) relations that can be learned from data and are widely used to infer the graphical representation of causal mechanisms [35, 6, 10, 4, 19, 18], not all CSI relations can be learned from mere observational data. However, several approaches exist for learning certain CSIs from data [21, 16, 32, 23]. In this paper, we investigate how CSIs allow for imitability in previously non-imitable instances.

After presenting the notation (Section 2), we first prove a hardness result, namely, that deciding the feasibility of imitation learning while accounting for CSIs is NP-hard. This is in contrast to the classic imitability problem [43], where deciding imitability is equivalent to a single d-separation test. We characterize a necessary and sufficient graphical criterion for imitability under CSIs under a structural assumption (Section 3) by leveraging a connection we establish between our problem and classic imitability. Next, we show that in certain instances, the dataset might allow for imitation, despite the fact that the graphical criterion is not satisfied (Section 4). Given the constructive nature of our achievability results, we propose algorithmic approaches for designing optimal imitation policies (Sections 3 and 4) and evaluate their performance in Section 5. The proofs of all our results appear in Appendix A.

## 2   Preliminaries

Throughout this work, we denote random variables and their realizations by capital and small letters, respectively. Likewise, we use boldface capital and small letters to represent sets of random variables and their realizations, respectively. For a variable $X$, $\mathcal{D}_X$ denotes the domain of $X$ and $\mathcal{P}_X$ the space of probability distributions over $\mathcal{D}_X$. Given two subsets of variables $\mathbf{T}$ and $\mathbf{S}$ such that $\mathbf{T} \subseteq \mathbf{S}$, and a realization $\mathbf{s} \in \mathcal{D}_\mathbf{S}$, we use $(\mathbf{s})_\mathbf{T}$ to denote the restriction of $\mathbf{s}$ to the variables in $\mathbf{T}$.

We use structural causal models (SCMs) as the semantic framework of our work [28]. An SCM $M$ is a tuple $\langle \mathbf{U}, \mathbf{V}, P^M(\mathbf{U}), \mathcal{F} \rangle$ where $\mathbf{U}$ and $\mathbf{V}$ are the sets of exogenous and endogenous variables, respectively. Values of variables in $\mathbf{U}$ are determined by an exogenous distribution $P^M(\mathbf{U})$, whereas the variables in $\mathbf{V}$ take values defined by the set of functions $\mathbf{F} = \{f_V^M\}_{V \in \mathbf{V}}$. That is, $V \leftarrow f_V^M(\mathbf{Pa}(V) \cup \mathbf{U}_V)$ where $\mathbf{Pa}(V) \subseteq \mathbf{V}$ and $\mathbf{U}_V \subseteq \mathbf{U}$. We also partition $\mathbf{V}$ into the observable and latent variables, denoted by $\mathbf{O}$ and $\mathbf{L}$, respectively, such that $\mathbf{V} = \mathbf{O} \cup \mathbf{L}$. The SCM induces a probability distribution over $\mathbf{V}$ whose marginal distribution over observable variables, denoted by $P^M(\mathbf{O})$, is called the *observational distribution*. Moreover, we use $do(\mathbf{X} = \mathbf{x})$ to denote an intervention on $\mathbf{X} \subseteq \mathbf{V}$ where the values of $\mathbf{X}$ are set to a constant $\mathbf{x}$ in lieu of the functions $\{f_X^M : \forall X \in \mathbf{X}\}$. We use $P_\mathbf{x}^M(\mathbf{y}) := P^M(\mathbf{Y} = \mathbf{y}|do(\mathbf{X} = \mathbf{x}))$ as a shorthand for the post-interventional distribution of $\mathbf{Y}$ after the intervention $do(\mathbf{X} = \mathbf{x})$.

Let $\mathbf{X}$, $\mathbf{Y}$, $\mathbf{W}$, and $\mathbf{Z}$ be pairwise disjoint subsets of variables. $\mathbf{X}$ and $\mathbf{Y}$ are called *contextually independent* given $\mathbf{W}$ in the context $\mathbf{z} \in \mathcal{D}_\mathbf{Z}$ if $P(\mathbf{X}|\mathbf{Y}, \mathbf{W}, \mathbf{z}) = P(\mathbf{X}|\mathbf{W}, \mathbf{z})$, whenever $P(\mathbf{Y}, \mathbf{W}, \mathbf{z}) > 0$. We denote this context-specific independence (CSI) relation by $\mathbf{X} \perp\!\!\!\perp \mathbf{Y}|\mathbf{W}, \mathbf{z}$. Moreover, a CSI is called *local* if it is of the form $X \perp\!\!\!\perp Y|\mathbf{z}$ where $\{Y\} \cup \mathbf{Z} \subseteq \mathbf{Pa}(X)$.

A directed acyclic graph (DAG) is defined as a pair $\mathcal{G} = (\mathbf{V}, \mathbf{E})$, where $\mathbf{V}$ is the set of vertices, and $\mathbf{E} \subseteq \mathbf{V} \times \mathbf{V}$ denotes the set of directed edges among the vertices. SCMs are associated with DAGs, where each variable is represented by a vertex of the DAG, and there is a directed edge from $V_i$ to $V_j$ if $V_i \in \mathbf{Pa}(V_j)$ [28]. Whenever a local CSI of the form $V_i \perp\!\!\!\perp V_j|\ell$ holds, we say $\ell$ is a label for the edge $(V_i, V_j)$ and denote it by $\ell \in \mathcal{L}_{(V_i, V_j)}$. Recalling the example of Figure 1c, the realization $T = 1$ is a label for the edge $(S, X)$, which indicates that this edge is absent when $T$ is equal to 1. Analogous to [29], we define a labeled DAG (LDAG), denoted by $\mathcal{G}^\mathcal{L}$, as a tuple $\mathcal{G}^\mathcal{L} = (\mathbf{V}, \mathbf{E}, \mathcal{L})$, where $\mathcal{L}$ denotes the labels representing local independence relations. More precisely, $\mathcal{L} = \{\mathcal{L}_{(V_i, V_j)} : \mathcal{L}_{(V_i, V_j)} \neq \emptyset \mid (V_i, V_j) \in \mathbf{E}\}$, where

$$\mathcal{L}_{(V_i, V_j)} = \{\ell \in \mathcal{D}_{\mathbf{V}'}|\mathbf{V}' \subseteq \mathbf{Pa}(V_j) \setminus \{V_i\}, V_i \perp\!\!\!\perp V_j|\ell\}.$$

Note that, when $\mathcal{L} = \emptyset$, $\mathcal{G}^\mathcal{L}$ reduces to a DAG. That is, every DAG is a special LDAG with no labels. For ease of notation, we drop the superscript $\mathcal{L}$ when $\mathcal{L} = \emptyset$. Given a label set $\mathcal{L}$, we define the context variables of $\mathcal{L}$, denoted by $\mathbf{C}(\mathcal{L})$, as the subset of variables that at least one realization of them appears in the edge labels. More precisely,

$$\mathbf{C}(\mathcal{L}) = \left\{V_i | \exists V_j, V_k \neq V_i, \mathbf{V}', \ell : (V_k, V_j) \in \mathbf{E}, V_i \in \mathbf{V}', \ell \in \mathcal{D}_{\mathbf{V}'} \cap \mathcal{L}_{(V_k, V_j)}\right\}, \tag{1}$$

where $\mathbf{V}'$ is an arbitrary subset of nodes containing $V_i$. The argument is that if there exists some arbitrary subset $\mathbf{V}'$ of nodes containing $V_i$ such that a realization $l$ of this subset results in independence (e.g., of some $V_j$ and $V_k$), then $V_i$ is considered as a context variable. We mainly focus on the settings where context variables are discrete or categorical. This assumption can be relaxed under certain considerations (See Remark 3.11). We let $\mathcal{M}_{\langle \mathcal{G}^{\mathcal{L}} \rangle}$ denote the class of SCMs compatible with the causal graph $\mathcal{G}^{\mathcal{L}}$. For a DAG $\mathcal{G}$, we use $\mathcal{G}_{\overline{\mathbf{X}}}$ and $\mathcal{G}_{\underline{\mathbf{X}}}$ to represent the subgraphs of $\mathcal{G}$ obtained by removing edges incoming to and outgoing from vertices of $\mathbf{X}$, respectively. We also use standard kin abbreviations to represent graphical relationships: the sets of parents, children, ancestors, and descendants of $\mathbf{X}$ in $\mathcal{G}$ are denoted by $\mathbf{Pa}(\mathbf{X})$ $\mathbf{Ch}(\mathbf{X})$, $\mathbf{An}(\mathbf{X})$, and $\mathbf{De}(\mathbf{X})$, respectively. For disjoint subsets of variable $\mathbf{X}$, $\mathbf{Y}$ and $\mathbf{Z}$ in $\mathcal{G}$, $\mathbf{X}$ and $\mathbf{Y}$ are said to be d-separated by $\mathbf{Z}$ in $\mathcal{G}$, denoted by $\mathbf{X} \perp \mathbf{Y}|\mathbf{Z}$, if every path between vertices in $\mathbf{X}$ and $\mathbf{Y}$ is blocked by $\mathbf{Z}$ (See Definition 1.2.3. in 28). Finally, solid and dashed vertices in the figures represent the observable and latent variables, respectively.

## 3 Imitability

In this section, we address the decision problem, i.e., whether imitation learning is feasible, given a causal mechanism. We first review the imitation learning problem from a causal perspective, analogous to the framework developed by [43]. We will use this framework to formalize the causal imitability problem in the presence of CSIs. Recall that $\mathbf{O}$ and $\mathbf{L}$ represented the observed and unobserved variables, respectively. We denote the action and reward variables by $X \in \mathbf{O}$ and $Y \in \mathbf{L}$, respectively. The reward variable is commonly assumed to be unobserved in imitation learning. Given a set of observable variables $\mathbf{Pa}^{\Pi} \subseteq \mathbf{O} \setminus \mathbf{De}(X)$, a policy $\pi$ is then defined as a stochastic mapping, denoted by $\pi(X|\mathbf{Pa}^{\Pi})$, mapping the values of $\mathbf{Pa}^{\Pi}$ to a probability distribution over the action $X$. Given a policy $\pi$, we use $do(\pi)$ to denote the intervention following the policy $\pi$, i.e., replacing the original function $f_X$ in the SCM by the stochastic mapping $\pi$. The distribution of variables under policy $do(\pi)$ can be expressed in terms of post-interventional distributions ($P(\cdot|do(x))$) as follows:

$$P(\mathbf{v}|do(\pi)) = \sum_{x \in \mathcal{D}_X, \mathbf{pa}^{\Pi} \in \mathcal{D}_{\mathbf{Pa}^{\Pi}}} P(\mathbf{v}|do(x), \mathbf{pa}^{\Pi})\pi(x|\mathbf{pa}^{\Pi})P(\mathbf{pa}^{\Pi}), \tag{2}$$

where $\mathbf{v}$ is a realization of an arbitrary subset $\mathbf{V}' \subseteq \mathbf{V}$. We refer to the collection of all possible policies as the *policy space*, denoted by $\Pi = \{\pi : \mathcal{D}_{\mathbf{Pa}^{\Pi}} \to \mathcal{P}_X\}$. Imitation learning is concerned with learning an optimal policy $\pi^* \in \Pi$ such that the reward distribution under policy $\pi^*$ matches that of the expert policy, that is, $P(y|do(\pi^*)) = P(y)$ [43]. Given a DAG $\mathcal{G}$ and the policy space $\Pi$, if such a policy exists, the instance is said to be *imitable* w.r.t. $\langle \mathcal{G}^{\mathcal{L}}, \Pi \rangle$. The causal imitability problem is formally defined below.

**Definition 3.1** (Classic imitability w.r.t. $\langle \mathcal{G}, \Pi \rangle$ 43)**.** *Given a latent DAG $\mathcal{G}$ and a policy space $\Pi$, let $Y$ be an arbitrary variable in $\mathcal{G}$. $P(y)$ is said to be imitable w.r.t. $\langle \mathcal{G}, \Pi \rangle$ if for any $M \in \mathcal{M}_{\langle \mathcal{G} \rangle}$, there exists a policy $\pi \in \Pi$ uniquely computable from $P(\mathbf{O})$ such that $P^M(y|do(\pi)) = P^M(y)$.*

Note that if $Y \notin \mathbf{De}(X)$, the third rule of Pearl's do calculus implies $P(y|do(x), \mathbf{pa}^{\Pi}) = P(y|\mathbf{pa}^{\Pi})$, and from Equation (2), $P(y|do(\pi)) = P(y)$ for any arbitrary policy $\pi$. Intuitively, in such a case, action $X$ has no effect on the reward $Y$, and regardless of the chosen policy, imitation is guaranteed. Therefore, throughout this work, we assume that $X$ affects $Y$, i.e., $Y \in \mathbf{De}(X) \cap \mathbf{L}$. Under this assumption, [43] proved that $P(y)$ is imitable w.r.t. $\langle \mathcal{G}, \Pi \rangle$ if and only if there exists a $\pi$-backdoor admissible set $\mathbf{Z}$ w.r.t. $\langle \mathcal{G}, \Pi \rangle$.

**Definition 3.2** ($\pi$-backdoor, 43)**.** *Given a DAG $\mathcal{G}$ and a policy space $\Pi$, a set $\mathbf{Z}$ is called $\pi$-backdoor admissible set w.r.t. $\langle \mathcal{G}, \Pi \rangle$ if and only if $\mathbf{Z} \subseteq \mathbf{Pa}^{\Pi}$ and $Y \perp X|\mathbf{Z}$ in $\mathcal{G}_{\underline{X}}$.*

The following lemma further reduces the search space of $\pi$-backdoor admissible sets to a single set.

**Lemma 3.3.** *Given a latent DAG $\mathcal{G}$ and a policy space $\Pi$, if there exists a $\pi$-backdoor admissible set w.r.t. $\langle \mathcal{G}, \Pi \rangle$, then $\mathbf{Z} = \mathbf{An}(\{X, Y\}) \cap (\mathbf{Pa}^{\Pi})$ is a $\pi$-backdoor admissible set w.r.t. $\langle \mathcal{G}, \Pi \rangle$.*

As a result, deciding the imitability reduces to testing a d-separation, i.e., whether $\mathbf{Z}$ defined in Lemma 3.3 d-separates $X$ and $Y$ in $\mathcal{G}_{\underline{X}}$, for which efficient algorithms exist [14, 38]. If this d-separation holds, then $\pi(X|\mathbf{Z}) = P(X|\mathbf{Z})$ is an optimal imitating policy. Otherwise, the instance is not imitable.

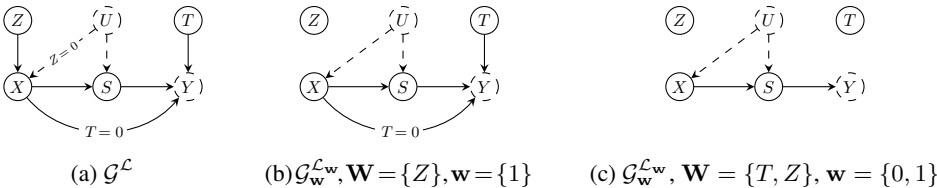

(a) $\mathcal{G}^{\mathcal{L}}$      (b) $\mathcal{G}_{\mathbf{w}}^{\mathcal{L}_{\mathbf{w}}}, \mathbf{W} = \{Z\}, \mathbf{w} = \{1\}$      (c) $\mathcal{G}_{\mathbf{w}}^{\mathcal{L}_{\mathbf{w}}}, \mathbf{W} = \{T, Z\}, \mathbf{w} = \{0, 1\}$

Figure 2: Two examples of context-induced subgraphs (Definition 3.5).

It is noteworthy that in practice, it is desirable to choose $\pi$-admissible sets with small cardinality for statistical efficiency. Polynomial-time algorithms for finding minimum(-cost) d-separators exist [2, 38].

## 3.1 Imitability with CSIs

Deciding the imitability when accounting for CSIs is not as straightforward as the classic case discussed earlier. In particular, as we shall see, the existence of $\pi$-backdoor admissible sets is not necessary to determine the imitability of $P(y)$ anymore in the presence of CSIs. In this section, we establish a connection between the classic imitability problem and imitability under CSIs. We begin with a formal definition of imitability in our setting.

**Definition 3.4** (Imitability w.r.t. $\langle \mathcal{G}^{\mathcal{L}}, \Pi \rangle$). *Given an LDAG $\mathcal{G}^{\mathcal{L}}$ and a policy space $\Pi$, let $Y$ be an arbitrary variable in $\mathcal{G}^{\mathcal{L}}$. $P(y)$ is called imitable w.r.t. $\langle \mathcal{G}^{\mathcal{L}}, \Pi \rangle$ if for any $M \in \mathcal{M}_{\langle \mathcal{G}^{\mathcal{L}} \rangle}$, there exists a policy $\pi \in \Pi$ uniquely computable from $P(\mathbf{O})$ such that $P^M(y|do(\pi)) = P^M(y)$.*

For an LDAG $\mathcal{G}^{\mathcal{L}}$, recall that we defined the set of context variables $\mathbf{C}(\mathcal{L})$ by Equation (1). The following definition is central in linking the imitability under CSIs to the classic case.

**Definition 3.5** (Context-induced subgraph). *Given an LDAG $\mathcal{G}^{\mathcal{L}} = (\mathbf{V}, \mathbf{E}, \mathcal{L})$, for a subset $\mathbf{W} \subseteq \mathbf{C}(\mathcal{L})$ and its realization $\mathbf{w} \in \mathcal{D}_{\mathbf{W}}$, we define the context-induced subgraph of $\mathcal{G}^{\mathcal{L}}$ w.r.t. $\mathbf{w}$, denoted by $\mathcal{G}_{\mathbf{w}}^{\mathcal{L}_{\mathbf{w}}}$, as the LDAG obtained from $\mathcal{G}^{\mathcal{L}}$ by keeping only the labels that are compatible with $\mathbf{w}$ [3], and deleting the edges that are absent given $\mathbf{W} = \mathbf{w}$, along with the edges incident to $\mathbf{W}$.*

Consider the example of Figure 2 for visualization. In the context $Z = 1$, the label $Z = 0$ on the edge $U \to X$ is discarded, as $Z = 0$ is not compatible with the context (see Figure 2b.) Note that edges incident to the context variable $Z$ are also omitted. On the other hand, in the context $Z = 1, T = 0$, the edge $X \to Y$ is absent and can be deleted from the corresponding graph (refer to Figure 2c.) Edges incident to both $T$ and $Z$ are removed in this case. Equipped with this definition, the following result, the proof of which appears in Appendix A, characterizes a necessary condition for imitability under CSIs.

**Lemma 3.6.** *Given an LDAG $\mathcal{G}^{\mathcal{L}}$ and a policy space $\Pi$, let $Y$ be an arbitrary variable in $\mathcal{G}^{\mathcal{L}}$. $P(y)$ is imitable w.r.t. $\langle \mathcal{G}^{\mathcal{L}}, \Pi \rangle$ only if $P(y)$ is imitable w.r.t. $\langle \mathcal{G}_{\mathbf{w}}^{\mathcal{L}_{\mathbf{w}}}, \Pi \rangle$ for every realization $\mathbf{w} \in \mathcal{D}_{\mathbf{w}}$ of every subset of variables $\mathbf{W} \subseteq \mathbf{C}(\mathcal{L})$.*

For instance, a necessary condition for the imitability of $P(y)$ in the graph of Figure 2a is that $P(y)$ is imitable in both 2b and 2c. Consider the following special case of Lemma 3.6: if $\mathbf{W} = \mathbf{C}(\mathcal{L})$, then $\mathcal{G}_{\mathbf{w}}^{\mathcal{L}_{\mathbf{w}}} = \mathcal{G}_{\mathbf{w}}$ is a DAG, as $\mathcal{L}_{\mathbf{w}} = \emptyset$ for every $\mathbf{w} \in \mathcal{D}_{\mathbf{W}}$. In essence, a necessary condition of imitability under CSIs can be expressed in terms of several classic imitability instances:

**Corollary 3.7.** *Given an LDAG $\mathcal{G}^{\mathcal{L}}$ and a policy space $\Pi$, let $Y$ be an arbitrary variable in $\mathcal{G}^{\mathcal{L}}$. $P(y)$ is imitable w.r.t. $\langle \mathcal{G}^{\mathcal{L}}, \Pi \rangle$ only if $P(y)$ is imitable w.r.t. $\langle \mathcal{G}_{\mathbf{c}}, \Pi \rangle$, i.e., there exists a $\pi$-backdoor admissible set w.r.t. $\langle \mathcal{G}_{\mathbf{c}}, \Pi \rangle$, for every $\mathbf{c} \in \mathcal{D}_{\mathbf{C}(\mathcal{L})}$.*

It is noteworthy that although the subgraphs $\mathcal{G}_{\mathbf{c}}$ in Corollary 3.7 are defined in terms of realizations of $\mathbf{C}(\mathcal{L})$, the number of such subgraphs does not exceed $2^{|\mathbf{E}|}$. This is due to the fact that $\mathcal{G}_{\mathbf{c}}$s share the same set of vertices, and their edges are subsets of the edges of $\mathcal{G}^{\mathcal{L}}$.

---

[3]Formally, we say that a label $l$ is compatible with a realization $w$ if they are consistent; in the sense that the variables at the intersection of the label and the realization take on the same values under both assignments.

Although deciding the classic imitability is straightforward, the number of instances in Corollary 3.7 can grow exponentially in the worst case. However, in view of the following hardness result, a more efficient criterion in terms of computational complexity cannot be expected.

**Theorem 3.8.** *Given an LDAG $\mathcal{G}^{\mathcal{L}}$ and a policy space $\Pi$, deciding the imitability of $P(y)$ w.r.t. $\langle \mathcal{G}^{\mathcal{L}}, \Pi \rangle$ is NP-hard.*

This result places the problem of causal imitability under CSI relations among the class of NP-hard problems in the field of causality, alongside other challenges such as devising minimum-cost interventions for query identification [3] and discovering the most plausible graph for causal effect identifiability [5]. Although Theorem 3.8 indicates that determining imitability under CSIs might be intractable in general, as we shall see in the next section taking into account only a handful of CSI relations can render previously non-imitable instances imitable. Before concluding this section, we consider a special yet important case of the general problem. Specifically, for the remainder of this section, we assume that $\mathbf{Pa}(\mathbf{C}(\mathcal{L})) \subseteq \mathbf{C}(\mathcal{L})$. That is, the context variables have parents only among the context variables. Under this assumption, the necessary criterion of Corollary 3.7 turns out to be sufficient for imitability as well. More precisely, we have the following characterization.

**Proposition 3.9.** *Given an LDAG $\mathcal{G}^{\mathcal{L}}$ where $\mathbf{Pa}(\mathbf{C}(\mathcal{L})) \subseteq \mathbf{C}(\mathcal{L})$ and a policy space $\Pi$, let $Y$ be an arbitrary variable in $\mathcal{G}^{\mathcal{L}}$. $P(y)$ is imitable w.r.t. $\langle \mathcal{G}^{\mathcal{L}}, \Pi \rangle$ if and only if $P(y)$ is imitable w.r.t. $\langle \mathcal{G}_{\mathbf{c}}, \Pi \rangle$, for every $\mathbf{c} \in \mathcal{D}_{\mathbf{C}(\mathcal{L})}$.*

---

**Algorithm 1** Imitation w.r.t. $\langle \mathcal{G}^{\mathcal{L}}, \Pi \rangle$

---

1: **function** IMITATE1 $(\mathcal{G}^{\mathcal{L}}, \Pi, X, Y)$
2:      Compute $\mathbf{C} := \mathbf{C}(\mathcal{L})$ using Equation (1)
3:      **for** $\mathbf{c} \in \mathcal{D}_{\mathbf{C}}$ **do**
4:          Construct a DAG $\mathcal{G}_{\mathbf{c}}$ using Definition 3.5
5:          **if** FINDSEP $(\mathcal{G}_{\mathbf{c}}, \Pi, X, Y)$ Fails **then**
6:              **return** FAIL
7:          **else**
8:              $\mathbf{Z}_{\mathbf{c}} \leftarrow$ FINDSEP $(\mathcal{G}_{\mathbf{c}}, \Pi, X, Y)$
9:              $\pi_{\mathbf{c}}(X|\mathbf{pa}^{\Pi}) \leftarrow P(X|(\mathbf{pa}^{\Pi})_{\mathbf{z}_{\mathbf{c}}})$
10:     $\pi^*(X|\mathbf{pa}^{\Pi}) \leftarrow \sum_{\mathbf{c} \in \mathcal{D}_{\mathbf{C}}} \mathbb{1}\{(\mathbf{pa}^{\Pi})_{\mathbf{C}} = \mathbf{c}\} \pi_{\mathbf{c}}(X|\mathbf{pa}^{\Pi})$
11:     **return** $\pi^*$

---

The proof of sufficiency, which is constructive, appears in Appendix A. The key insight here is that an optimal imitation policy is constructed based on the imitation policies corresponding to the instances $\langle \mathcal{G}_{\mathbf{c}}, \Pi \rangle$. In view of proposition 3.9, we provide an algorithmic approach for finding an optimal imitation policy under CSIs, as described in Algorithm 1. This algorithm takes as input an LDAG $\mathcal{G}^{\mathcal{L}}$, a policy space $\Pi$, an action variable $X$ and a latent reward $Y$. It begins with identifying the context variables $\mathbf{C}(= \mathbf{C}(\mathcal{L}))$, defined by Equation (1) (Line 2). Next, for each realization $\mathbf{c} \in \mathcal{D}_{\mathbf{C}}$, the corresponding context-induced subgraph (Def. 3.5) is built (which is a DAG). If $P(y)$ is not imitable in any of these DAGs, the algorithm fails, i.e., declares $P(y)$ is not imitable in $\mathcal{G}^{\mathcal{L}}$. The imitability in each DAG is checked through a d-separation based on Lemma 3.3 (for further details of the function $FindSep$, see Algorithm 3 in Appendix B.) Otherwise, for each realization $\mathbf{c} \in \mathcal{D}_{\mathbf{C}}$, an optimal policy $\pi_c$ is learned through the application of the $\pi$-backdoor admissible set criterion (line 9). If such a policy exists for every realization of $\mathbf{C}$, $P(y)$ is imitable w.r.t. $\langle \mathcal{G}^{\mathcal{L}}, \Pi \rangle$ due to Proposition 3.9. An optimal imitating policy $\pi^*$ is computed based on the previously identified policies $\pi_{\mathbf{c}}$. Specifically, $\pi^*$, the output of the algorithm, is defined as

$$\pi^*(X|\mathbf{pa}^{\Pi}) \leftarrow \sum_{\mathbf{c} \in \mathcal{D}_{\mathbf{C}}} \mathbb{1}\{(\mathbf{pa}^{\Pi})_{\mathbf{C}} = \mathbf{c}\} \pi_{\mathbf{c}}(X|\mathbf{pa}^{\Pi}).$$

**Theorem 3.10.** *Algorithm 1 is sound and complete for determining the imitability of $P(y)$ w.r.t. $\langle \mathcal{G}^{\mathcal{L}}, \Pi \rangle$ and finding the optimal imitating policy in the imitable case, under the assumption that $\mathbf{Pa}(\mathbf{C}(\mathcal{L})) \subseteq \mathbf{C}(\mathcal{L})$.*

**Remark 3.11.** *Algorithm 1 requires assessing the d-separation of line 5 in the context-induced subgraphs $\mathcal{G}_{\mathbf{c}}$. Even when the context variables $\mathbf{C}$ are continuous, the domain of these variables can*

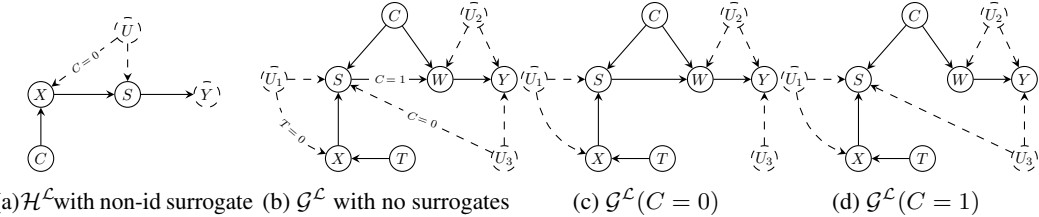

(a)$\mathcal{H}^{\mathcal{L}}$with non-id surrogate (b) $\mathcal{G}^{\mathcal{L}}$ with no surrogates     (c) $\mathcal{G}^{\mathcal{L}}(C = 0)$     (d) $\mathcal{G}^{\mathcal{L}}(C = 1)$

Figure 3: Two examples of leveraging CSI relations to achieve imitability.

*be partitioned into at most $2^m$ equivalence classes in terms of their corresponding context-induced subgraph, where $m$ denotes the number of labeled edges. This holds since the number of context-induced subgraphs cannot exceed $2^m$. It is noteworthy, however, that solving the equation referred to in line 10 of Algorithm 2 for continuous variables may bring additional computational challenges.*

**Remark 3.12.** *Under certain assumptions, polynomial-time algorithms can be devised for deciding imitability. One such instance is when $\mathcal{D}_{\mathbf{C}(\mathcal{L})} = \mathcal{O}\big(\log(|V|)\big)$. Another analogous case is when the number of context-specific edges is $\mathcal{O}\big(\log(|V|)\big)$. Both of these lead polynomially many context-induced subgraphs in terms of $|V|$, which in turn implies that Alg. 1 runs in polynomial time.*

## 4 Leveraging causal effect identifiability for causal imitation learning

Arguably, the main challenge in imitation learning stems from the latent reward. However, in certain cases, there exist observable variables $\mathbf{S} \subseteq \mathbf{O}$ such that $P(\mathbf{S}|do(\pi)) = P(\mathbf{S})$ implies $P(y|do(\pi)) = P(y)$ for any policy $\pi \in \Pi$. Such $\mathbf{S}$ is said to be an imitation surrogate for $Y$ [43]. Consider, for instance, the graph of Figure 3a, where $X$ represents the pricing strategy of a company, $C$ is a binary variable indicating recession ($C = 0$) or expansion ($C = 1$) period, $U$ denotes factors such as demand and competition in the market, $S$ represents the sales and $Y$ is the overall profit of the company. Due to Proposition 3.9, $P(y)$ is not imitable in this graph. On the other hand, the sales figure ($S$) is an imitation surrogate for the profit ($Y$), as it can be shown that whenever $P(S|do(\pi)) = P(S)$ for a given policy $\pi$, $P(y|do(\pi)) = P(y)$ holds for the same policy. Yet, according to do-calculus, $P(S|do(\pi))$ itself is not identifiable due to the common confounding $U$. On the other hand, we note that the company's pricing strategy ($X$) becomes independent of demand ($U$) during a recession ($C = 0$), as the company may not have enough customers regardless of the price it sets. This CSI relation amounts to the following identification formula for the effect of an arbitrary policy $\pi$ on sales figures:

$$P(s|do(\pi)) = \sum_{x,c} P(s|x, C = 0)\pi(x|c)P(c), \tag{3}$$

where all of the terms on the right-hand side are known given the observations. Note that even though $S$ is a surrogate in Figure 3a, without the CSI $C = 0$, we could not have written Equation (3). Given the identification result of this equation, if the set of equations $P(s|do(\pi)) = P(s)$ has a solution $\pi^*$, then $\pi^*$ becomes an imitation policy for $P(y)$ as well [43]. It is noteworthy that solving the aforementioned linear system of equations for $\pi^*$ is straightforward, for it boils down to a matrix inversion[4]. In the example discussed, although the graphical criterion of Proposition 3.9 does not hold, the data-specific parameters could yield imitability in the case that these equations are solvable. We therefore say $P(y)$ is imitable w.r.t. $\langle \mathcal{G}^{\mathcal{L}}, \Pi, P(\mathbf{O})\rangle$, as opposed to 'w.r.t. $\langle \mathcal{G}^{\mathcal{L}}, \Pi \rangle$'. Precisely, we say $P(y)$ is imitable w.r.t. $\langle \mathcal{G}^{\mathcal{L}}, \Pi, P(\mathbf{O})\rangle$ if for every $M \in \mathcal{M}_{\langle \mathcal{G}^{\mathcal{L}}\rangle}$ such that $P^M(\mathbf{O}) = P(\mathbf{O})$, there exists a policy $\pi$ such that $P^M(y|do(\pi)) = P^M(y)$.

The idea of surrogates could turn out to be useful to circumvent the imitability problem when the graphical criterion does not hold. In Figure 3b, however, neither graphical criteria yields imitability nor any imitation surrogates exist. In what follows, we discuss how CSIs can help circumvent the problem of imitability in even in such instances. Given an LDAG $\mathcal{G}^{\mathcal{L}}$ and a context $\mathbf{C} = \mathbf{c}$, we denote the *context-specific* DAG w.r.t. $\langle \mathcal{G}^{\mathcal{L}}, \mathbf{c} \rangle$ by $\mathcal{G}^{\mathcal{L}}(\mathbf{c})$ where $\mathcal{G}^{\mathcal{L}}(\mathbf{c})$ is the DAG obtained by deleting all the spurious edges, i.e., the edges that are absent given the context $\mathbf{C} = \mathbf{c}$, from $\mathcal{G}^{\mathcal{L}}$.

---

[4]In the discrete case, and kernel inversion in the continuous case.

**Algorithm 2** Imitation w.r.t. $\langle \mathcal{G}^{\mathcal{L}}, \Pi, P(\mathbf{O})\rangle$

---

1: **function** IMITATE2 $(\mathcal{G}^{\mathcal{L}}, \Pi, X, Y, P(\mathbf{O}))$
2:     **if** SUBIMITATE $(\mathcal{G}^{\mathcal{L}}, \Pi, X, Y, \emptyset, \emptyset, P(\mathbf{O}))$ **then**
3:         $\pi^{*} \leftarrow$ SUBIMITATE $(\mathcal{G}^{\mathcal{L}}, \Pi, X, Y, \emptyset, \emptyset, P(\mathbf{O}))$
4:         **return** $\pi^{*}$
5:     **else**
6:         **return** Fail

---

1: **function** SUBIMITATE $(\mathcal{G}^{\mathcal{L}}, \Pi, X, Y, \mathbf{C}, \mathbf{c}, P(\mathbf{O}))$
2:     Construct the context-specific DAG $\mathcal{G}^{\mathcal{L}}(\mathbf{c})$
3:     **if** FINDSEP $(\mathcal{G}^{\mathcal{L}}(\mathbf{c}), \Pi, X, Y)$ **then**
4:         $\mathbf{Z_c} \leftarrow$ FINDSEP $(\mathcal{G}^{\mathcal{L}}(\mathbf{c}), \Pi, X, Y)$
5:         $\pi_{\mathbf{c}}(X | \mathbf{Pa}^{\Pi}) \leftarrow P(X | \mathbf{Z_c}, \mathbf{C} = \mathbf{c})$
6:         **return** $\pi_{\mathbf{c}}$
7:     **else if** FINDMINSEP $(\mathcal{G}^{\mathcal{L}}(\mathbf{c}) \cup \Pi, X, Y, \mathbf{C})$ **then**
8:         $\mathbf{S_c} \leftarrow$ FINDMINSEP $(\mathcal{G}^{\mathcal{L}}(\mathbf{c}) \cup \Pi, X, Y, \mathbf{C}) \backslash \mathbf{C}$
9:         **if** CSI-ID $(P(\mathbf{S_c} | do(x), \mathbf{C} = \mathbf{c}))$ **then**
10:            Solve $P(\mathbf{S_c} | do(\pi_{\mathbf{c}}), \mathbf{c}) = P(\mathbf{S_c} | \mathbf{c})$ for $\pi_{\mathbf{c}} \in \Pi$
11:            **if** such $\pi_{\mathbf{c}}$ exists **then**
12:                **return** $\pi_{\mathbf{c}}$
13:     **else**
14:         **if** $\mathbf{C} = \mathbf{C}(\mathcal{L}) \cap \mathbf{Pa}^{\Pi}$ **then**
15:            **return** FAIL
16:         **else**
17:            Choose $V \in \mathbf{C}(\mathcal{L}) \cap \mathbf{Pa}^{\Pi} \setminus \mathbf{C}$ arbitrarily
18:            **for** $v \in \mathcal{D}_V$ **do**
19:                $\mathbf{C}' \leftarrow \mathbf{C} \cup \{V\}$, $\mathbf{c}' \leftarrow \mathbf{c} \cup \{v\}$
20:                **if** SI$(\mathcal{G}^{\mathcal{L}}, \Pi, X, Y, \mathbf{C}', \mathbf{c}', P(\mathbf{O}))$ **then**
21:                    $\pi_{\mathbf{c},v} \leftarrow$ SI$(\mathcal{G}^{\mathcal{L}}, \Pi, X, Y, \mathbf{C}', \mathbf{c}', P(\mathbf{O}))$
22:                **else**
23:                    **return** Fail
24:            $\pi_{\mathbf{c}}(X | \mathbf{Pa}^{\Pi}) \leftarrow \sum_{v \in \mathcal{D}_V} \mathbb{1}_{\{(\mathbf{Pa}^{\Pi})_V = v\}} \pi_{\mathbf{c},v}$
25:            **return** $\pi_{\mathbf{c}}$

---

**Definition 4.1** (Context-specific surrogate). *Given an LDAG $\mathcal{G}^{\mathcal{L}}$, a policy space $\Pi$, and a context $\mathbf{c}$, a set of variables $\mathbf{S} \subseteq \mathbf{O}$ is called context-specific surrogate w.r.t. $\langle \mathcal{G}^{\mathcal{L}}(\mathbf{c}), \Pi \rangle$ if $X \perp Y | \{\mathbf{S}, \mathbf{C}\}$ in $\mathcal{G}^{\mathcal{L}}(\mathbf{c}) \cup \Pi$ where $\mathcal{G}^{\mathcal{L}}(\mathbf{c}) \cup \Pi$ is a supergraph of $\mathcal{G}^{\mathcal{L}}(\mathbf{c})$ by adding edges from $\mathbf{Pa}^{\Pi}$ to $X$.*

Consider the LDAG of Figure 3b for visualization. The context-specific DAGs corresponding to contexts $C = 0$ and $C = 1$ are shown in Figures 3c and 3d, respectively. $S$ is a context-specific surrogate with respect to $\langle \mathcal{G}^{\mathcal{L}}(C = 0), \Pi \rangle$, while no context-specific surrogates exist with respect to $\langle \mathcal{G}^{\mathcal{L}}(C = 1), \Pi \rangle$. The following result indicates that despite the absence of a general imitation surrogate, the existence of context-specific surrogates could suffice for imitation.

**Proposition 4.2.** *Given an LDAG $\mathcal{G}^{\mathcal{L}}$ and a policy space $\Pi$, let $\mathbf{C}$ be a subset of $\mathbf{Pa}^{\Pi} \cap \mathbf{C}(\mathcal{L})$. If for every realization $\mathbf{c} \in \mathcal{D}_{\mathbf{C}}$ at least one of the following holds, then $P(y)$ is imitable w.r.t. $\langle \mathcal{G}^{\mathcal{L}}, \Pi, P(\mathbf{O}) \rangle$.*

- *There exists a subset $\mathbf{S_c}$ such that $X \perp Y | \{\mathbf{S_c}, \mathbf{C}\}$ in $\mathcal{G}^{\mathcal{L}}(\mathbf{c}) \cup \Pi$, and $P(\mathbf{S_c} | do(\pi), \mathbf{C} = \mathbf{c}) = P(\mathbf{S_c} | \mathbf{C} = \mathbf{c})$ has a solution $\pi_{\mathbf{c}} \in \Pi$.*

- *There exists $\mathbf{Z}_c \subseteq \mathbf{Pa}^{\Pi}$ such that $X \perp Y | \{\mathbf{Z_c}, \mathbf{C}\}$ in $\mathcal{G}^{\mathcal{L}}(\mathbf{c})_{\underline{X}}$.*

As a concrete example, defining $\mathbf{Z}_1 = \emptyset$, $X \perp Y | \{\mathbf{Z}_1, C\}$ holds in the graph of Figure 3d. Moreover, $\mathbf{S}_0 = \{S\}$, is a context-specific surrogate w.r.t. $\langle \mathcal{G}^{\mathcal{L}}(C = 0), \Pi \rangle$. As a consequence of Proposition 4.2, $P(y)$ is imitable w.r.t. $\langle \mathcal{G}^{\mathcal{L}}, \Pi, P(\mathbf{O}) \rangle$, if a solution $\pi_0$ exists to the linear set of equations

$$P(S | do(\pi), C = 0) = P(S | C = 0),$$

where $P(s | do(\pi), C = 0) = \sum_{x,t} P(s | x, T = 0, C = 0) \pi(x | t, C = 0) P(t | C = 0)$, analogous to Equation (3).

Table 1: Results pertaining to the model of Figure 3a.

| Metric | Algorithm | | | |
|---|---|---|---|---|
| | Expert | Naive 1 | Naive 2 | Algorithm 2 |
| $\mathbb{E}[Y]$ | 1.367 | 1.194 | 1.193 | 1.358 |
| $D_{KL}(P(Y)\|P(Y\|do(\pi_{ALG}))$ | 0 | 0.0217 | 0.0219 | 0.0007 |
| $D_{KL}(\pi_{ALG}(X\|T=0)\|\hat{\pi}_{ALG}(X\|T=0))$ | $NA$ | $2.3 \times 10^{-5}$ | $4.4 \times 10^{-6}$ | $1.3 \times 10^{-3}$ |
| $D_{KL}(\pi_{ALG}(X\|T=1)\|\hat{\pi}_{ALG}(X\|T=1))$ | $NA$ | $2.3 \times 10^{-5}$ | $4.8 \times 10^{-4}$ | $1.3 \times 10^{-3}$ |

To sum up, accounting for CSIs has a two-fold benefit: (a) *context-specific surrogates* can be leveraged to render previously non-imitable instances imitable, and (b) identification results can be derived for imitation surrogates that were previously non-identifiable.

In light of Proposition 4.2, an algorithmic approach for causal imitation learning is proposed, summarized as Algorithm 2. This algorithm calls a recursive subroutine, $SubImitate$, also called $SI$ within the pseudo-code. It is noteworthy that Proposition 4.2 guarantees imitability if the two conditions are met for any arbitrary subset of $\mathbf{C}(\mathcal{L}) \cap \mathbf{Pa}^{\Pi}$. As we shall see, Algorithm 2 utilizes a recursive approach for building such a subset so as to circumvent the need to test all of the possibly exponentially many subsets.

The subroutine $SI$ is initiated with an empty set ($\mathbf{C} = \emptyset$) as the considered context variables at the first iteration. At each iteration, the realizations of $\mathbf{C}$ are treated separately. For each such realization $\mathbf{c}$, if the second condition of Proposition 4.2 is met through a set $\mathbf{Z_c}$, then $P(X|\mathbf{Z_c}, \mathbf{C} = \mathbf{c})$ is returned as the context-specific imitating policy (lines 3-6). Otherwise, the search for a context-specific surrogate begins. We utilize the $FindMinSep$ algorithm of [40] to identify a minimal separating set $\mathbf{S_c} \cup \mathbf{C}$ for $X$ and $Y$, among those that necessarily include $\mathbf{C}$ (lines 7-8). We then use the identification algorithm of [39] under CSI relations to identify the effect of an arbitrary policy on $\mathbf{S_c}$, conditioned on the context $\mathbf{c}$. This algorithm is built upon CSI-calculus, which subsumes do-calculus[5]. Next, if the linear system of equations $P(\mathbf{S_c}|do(\pi_{\mathbf{c}}), \mathbf{c}) = P(\mathbf{S_c}|, \mathbf{c})$ has a solution, then this solution is returned as the optimal policy (lines 9-12). Otherwise, an arbitrary variable $V \in \mathbf{C}(\mathcal{L}) \cap \mathbf{Pa}^{\Pi} \setminus \mathbf{C}$ is added to the considered context variables, and the search for context-specific policies proceeds while taking the realizations of $V$ into account (lines 17-24). If no variables are left to add to the set of context variables (i.e., $\mathbf{C} = \mathbf{C}(\mathcal{L}) \cap \mathbf{Pa}^{\Pi}$) and neither of the conditions of Proposition 4.2 are met for a realization of $\mathbf{C}$, then the algorithm stops with a failure (lines 14-15). Otherwise, an imitating policy $\pi^*$ is returned. We finally note that if computational costs matter, the CSI-ID function of line 9 can be replaced by the ID algorithm of [33]. Further, the minimal separating sets of line 8 might not be unique, in which case all such sets can be used.

**Theorem 4.3.** *Given an LDAG $\mathcal{G}^{\mathcal{L}}$, a policy space $\Pi$ and observational distribution $P(\mathbf{O})$, if Algorithm 2 returns a policy $\pi^*$, then $\pi^*$ is an optimal imitating policy for $P(y)$ w.r.t. $\langle \mathcal{G}^{\mathcal{L}}, \Pi, P(\mathbf{O}) \rangle$. That is, Algorithm 2 is sound [6].*

## 5 Experiments

Our experimental evaluation is organized into two parts. In the first part, we address the decision problem pertaining to imitability. We evaluate the gain resulting from accounting for CSIs in rendering previously non-imitable instances imitable. In particular, we assess the classic imitability v.s. imitability under CSIs for randomly generated graphs. In the second part, we compare the performance of Alg. 2 against baseline algorithms on synthetic datasets (see Sec. D for further details of our experimental setup). Python implementation are accessible at `https://github.com/SinaAkbarii/causal-imitation-learning/`.

### 5.1 Evaluating imitability

We sampled random graphs with $n$ vertices and maximum degree $\Delta = \frac{n}{10}$ uniformly at random. Each variable was assumed to be latent with probability $\frac{1}{6}$. We chose 3 random context variables

---

[5]Figure 3a is an example where do-calculus fails to identify $P(s|do(x))$, whereas CSI-calculus provides an identification formula, Equation (3).

[6]See Section C for an example where Algorithm 2 is not complete.

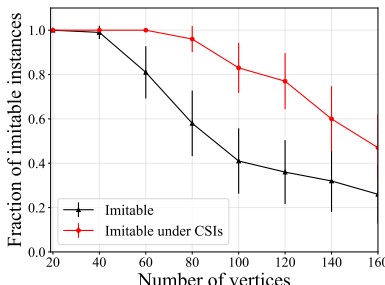

Figure 4: The fraction of imitable instances (in the classical sense) vs those that are imitable considering CSIs.

such that the graphical constraint $\mathbf{Pa}(\mathbf{C}(\mathcal{L})) \subseteq \mathbf{C}(\mathcal{L})$ is satisfied. Labels on the edges were sampled with probability $0.5$. We then evaluated the graphical criterion of classic imitability (existence of $\pi$-backdoor) and imitability with CSIs (Corollary 3.7). The results are depicted in Figure 4, where each point in the plot is an average of 100 sampled graphs. As seen in this figure, taking into account only a handful of CSI relations (in particular, 3 context variables among hundreds of variables) could significantly increase the fraction of imitable instances.

## 5.2    Performance evaluation

In this section, we considered the graph of Figure 3b as a generalization of the economic model of Figure 3a, where the reward variable $Y$ can be a more complex function. As discussed earlier, this graph has neither a $\pi$-backdoor admissible set nor an imitation surrogate. However, given the identifiability of $P(S|do(\pi), C = 0)$, Algorithm 2 can achieve an optimal policy. We compared the performance of the policy returned by Algorithm 2 against two baseline algorithms: Naive algorithm 1, which mimics only the observed distribution of the action variable (by choosing $\pi(X) = P(X)$), and Naive algorithm 2, which takes the causal ancestors of $X$ into account, designing the policy $\pi(X|T) = P(X|T)$. Naive algorithm 2 can be thought of as a feature selection followed by a behavior cloning approach. The goal of this experiment was to demonstrate the infeasibility of imitation learning without taking CSIs into account. A model with binary observable variables and a ternary reward was generated. Let $\pi_{ALG}$ represent the policy that the algorithm would have learned with infinite number of samples, and $\hat{\pi}_{ALG}$ the policy it learns with the given finite sample size. As can be seen in Table 1, Algorithm 2 was able to match the expert policy both in expected reward and KL divergence of the reward distribution. The naive algorithms, on the other hand, failed to get close to the reward distribution of the expert. Since the algorithms were fed with finite observational samples, the KL divergence of the estimated policies with the nominal policy is also reported. Notably, based on the reported measures, the undesirable performance of the Naive algorithms does not stem from estimation errors.

## 6    Concluding remarks

We considered the causal imitation learning problem when accounting for context-specific independence relations. We proved that in contrast to the classic problem, which is equivalent to a d-separation, the decision problem of imitability under CSIs is NP-hard. We established a link between these two problems. In particular, we proved that imitability under CSIs is equivalent to several instances of the classic imitability problem for a certain class of context variables. We showed that utilizing the overlooked notion of CSIs could be a worthwhile tool in causal imitation learning as an example of a fundamental AI problem from a causal perspective. We note that while taking a few CSI relations into account could result in significant achievable results, the theory of CSIs is not yet well-developed. In particular, there exists no complete algorithm for causal identification under CSIs. Further research on the theory of CSI relations could yield considerable benefits in various domains where such relations are present.

## Acknowledgements

This research was in part supported by the Swiss National Science Foundation under NCCR Automation, grant agreement 51NF40_180545 and Swiss SNF project 200021_204355 /1.

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

# Appendix

Proofs of the main results are given in Section A. The details of FindSep subroutine are presented in Section B. In Section C provides an example where Alg. 2 is not complete, and Section D includes the details of our experiment setup.

## A  Technical proofs

**Lemma 3.3.** *Given a latent DAG $\mathcal{G}$ and a policy space $\Pi$, if there exists a $\pi$-backdoor admissible set w.r.t. $\langle \mathcal{G}, \Pi \rangle$, then $\mathbf{Z} = \mathbf{An}(\{X, Y\}) \cap (\mathbf{Pa}^{\Pi})$ is a $\pi$-backdoor admissible set w.r.t. $\langle \mathcal{G}, \Pi \rangle$.*

*Proof.* We first claim that for any set $\mathbf{W} \subseteq \{X, Y\}$, the set of ancestors of $\mathbf{W}$ in $\mathcal{G}$ and $\mathcal{G}_{\underline{X}}$ coincide, i.e., $\mathbf{An}(\mathbf{W})_{\mathcal{G}} = An(\mathbf{W})_{\mathcal{G}_{\underline{X}}}$. First, note that since $\mathcal{G}_{\underline{X}}$ is a subgraph of $\mathbf{G}$, $An(\mathbf{W})_{\mathcal{G}_{\underline{X}}} \subseteq An(\mathbf{W})_{\mathcal{G}}$. For the other direction, let $T \in An(\mathbf{W})_{\mathcal{G}}$ be an arbitrary vertex. If $T \in An(X)$, then $T \in An(\mathbf{W})_{\mathcal{G}_{\underline{X}}}$. Otherwise, there exists a directed path from $T$ to $Y$ that does not pass through $X$. The same directed path exists in $\mathcal{G}_{\underline{X}}$, and as a result, $T \in An(\mathbf{W})_{\mathcal{G}_{\underline{X}}}$. Therefore, $An(\mathbf{W})_{\mathcal{G}} \subseteq An(\mathbf{W})_{\mathcal{G}_{\underline{X}}}$.

The rest of the proof follows from an application of Lemma 3.4 of [40] in $\mathcal{G}_{\underline{X}}$, with $I = \emptyset$ and $R = \mathbf{Pa}^{\Pi}$. $\qquad\square$

**Lemma 3.6.** *Given an LDAG $\mathcal{G}^{\mathcal{L}}$ and a policy space $\Pi$, let $Y$ be an arbitrary variable in $\mathcal{G}^{\mathcal{L}}$. $P(y)$ is imitable w.r.t. $\langle \mathcal{G}^{\mathcal{L}}, \Pi \rangle$ only if $P(y)$ is imitable w.r.t. $\langle \mathcal{G}^{\mathcal{L}_{\mathbf{w}}}_{\mathbf{w}}, \Pi \rangle$ for every realization $\mathbf{w} \in \mathcal{D}_{\mathbf{w}}$ of every subset of variables $\mathbf{W} \subseteq \mathbf{C}(\mathcal{L})$.*

*Proof.* Assume the contrary, that there exists a subset of variables $\mathbf{W} \subseteq \mathbf{V} \setminus Y$ such that for an assignment $\mathbf{w}_0 \in \mathcal{D}_{\mathbf{W}}$, $P(y)$ is not imitable w.r.t. $\langle \mathcal{G}^{\mathcal{L}_{\mathbf{w}_0}}_{\mathbf{w}_0}, \Pi \rangle$. It suffices to show that $P(y)$ is not imitable w.r.t. $\langle \mathcal{G}^{\mathcal{L}}, \Pi \rangle$. Since $P(y)$ is not imitable w.r.t. $\langle \mathcal{G}^{\mathcal{L}_{\mathbf{w}_0}}_{\mathbf{w}_0}, \Pi \rangle$, there exists an SCM $M' \in \mathcal{M}_{\langle \mathcal{G}^{\mathcal{L}_{\mathbf{w}_0}}_{\mathbf{w}_0} \rangle}$ such that $P^{M'}(y|do(\pi)) \neq P^{M'}(y)$ for every $\pi \in \Pi$. Define

$$\epsilon := \min_{\pi \in \Pi} |P^{M'}(y) - P^{M'}(y|do(\pi))|. \tag{4}$$

Also, define

$$\delta := \min\{\frac{\epsilon}{4}, \frac{1}{2}\}. \tag{5}$$

Note that $\epsilon > 0$, and $0 < \delta < 1$. Next, construct an SCM $M$ over $\mathbf{V}$ compatible with $\mathcal{G}^{\mathcal{L}_{\mathbf{w}_0}}_{\mathbf{w}_0}$ as follows. For variables $\mathbf{W}$, $P^M(\mathbf{W} = \mathbf{w}_0) = 1 - \delta$, and the rest is uniformly distributed over other realizations (summing up to $\delta$), such that $P^M(\mathbf{W} \neq \mathbf{w}_0) = \delta$. For variables $V_i \in \mathbf{V} \setminus \mathbf{W}$, $P^M(V_i|\mathbf{Pa}(V_i)) = P^{M'}(V_i|\mathbf{Pa}(V_i))$, i.e., they follow the same law as the model $M'$. Note that by construction, $\mathbf{W}$ are isolated vertices in $\mathcal{G}^{\mathcal{L}_{\mathbf{w}_0}}_{\mathbf{w}_0}$, and therefore independent of every other variable in both $M'$ and $M$. Therefore,

$$P^M(y) = \sum_{\mathbf{w} \in \mathcal{D}_{\mathbf{W}}} P^M(y|\mathbf{w})P^M(\mathbf{w}) = \sum_{\mathbf{w} \in \mathcal{D}_{\mathbf{W}}} P^{M'}(y|\mathbf{w})P^M(\mathbf{w}) = \sum_{\mathbf{w} \in \mathcal{D}_{\mathbf{W}}} P^{M'}(y)P^M(\mathbf{w})$$
$$= P^{M'}(y). \tag{6}$$

Moreover, for any policy $\pi_1 \in \Pi$ dependant on the values of $\mathbf{Z}' \subseteq \mathbf{Pa}^{\Pi}$, define a policy $\pi_2 \in \Pi$ dependant on $\mathbf{Z} = \mathbf{Z}' \setminus \mathbf{W}$ as $\pi_2(X|\mathbf{Z}) = (1-\delta)\pi_1(X|\mathbf{Z}, \mathbf{W} = \mathbf{w}_0) + \delta\pi_1(X|\mathbf{Z}, \mathbf{W} \neq \mathbf{w}_0)$. Note

that $\mathbf{Z} \cap \mathbf{W} = \emptyset$. That is, $\pi_2$ is independent of the values of $\mathbf{W}$. We can write

$$
\begin{aligned}
P^M(y|do(\pi_1)) &= \sum_{\mathbf{w} \in \mathcal{D}_{\mathbf{W}}} P^M(y|do(\pi_1), \mathbf{w}) P^M(\mathbf{w}|do(\pi)) \\
&= P^M(y|do(\pi_1), \mathbf{W} = \mathbf{w}_0)(1 - \delta) + P^M(y|do(\pi_1), \mathbf{W} \neq \mathbf{w}_0)\delta \\
&= \sum_{x,\mathbf{z}} (1 - \delta) P^M(y|do(x), \mathbf{z}, \mathbf{W} = \mathbf{w}_0) \pi_1(x|\mathbf{z}, \mathbf{W} = \mathbf{w}_0) P^M(\mathbf{z}|\mathbf{W} = \mathbf{w}_0) \\
&+ \sum_{x,\mathbf{z}} \delta P^M(y|do(x), \mathbf{z}, \mathbf{W} \neq \mathbf{w}_0) \pi_1(x|\mathbf{z}, \mathbf{W} \neq \mathbf{w}_0) P^M(\mathbf{z}|\mathbf{W} \neq \mathbf{w}_0) \\
&= \sum_{x,\mathbf{z}} (1 - \delta) P^{M'}(y|do(x), \mathbf{z}) \pi_1(x|\mathbf{z}, \mathbf{W} = \mathbf{w}_0) P^{M'}(\mathbf{z}) \\
&+ \sum_{x,\mathbf{z}} \delta P^{M'}(y|do(x), \mathbf{z}) \pi_1(x|\mathbf{z}, \mathbf{W} \neq \mathbf{w}_0) P^{M'}(\mathbf{z}) \\
&= \sum_{x,\mathbf{z}} P^{M'}(y|do(x), \mathbf{z}) \pi_2(x|\mathbf{z}) P^{M'}(\mathbf{z}) \\
&= P^{M'}(y|do(\pi_2)).
\end{aligned}
\tag{7}
$$

That is, for any policy $\pi_1$ under model $M$, there is a policy $\pi_2$ that results in the same reward distribution under model $M'$. Therefore, combining Equations (6) and (7),

$$
\min_{\pi \in \Pi} |P^M(y) - P^M(y|do(\pi))| = \min_{\pi \in \Pi} |P^{M'}(y) - P^{M'}(y|do(\pi))|,
$$

although this minimum might occur under different policies. Recalling Equation (4), we get that under Model $M$ and for any policy $\pi \in \Pi$,

$$
\epsilon \leq |P^M(y) - P^M(y|do(\pi))|.
\tag{8}
$$

Now we construct yet another SCM $M^\delta$ over $\mathbf{V}$ as follows. Variables $\mathbf{W}$ are distributed as in model $M$, i.e., $P^{M^\delta}(\mathbf{W}) = P^M(\mathbf{W})$. Moreover, for each variable $V \in \mathbf{V} \setminus \mathbf{W}$ set $V = V^M$ if $\mathbf{Pa}(V) \cap \mathbf{W} = (\mathbf{w}_0)_{\mathbf{Pa}(V) \cap \mathbf{W}}$ where $V^M$ denotes the same variable $V$ under SCM $M$. Otherwise, distribution of $V$ is uniform in $\mathcal{D}_V$. Note that by definition of $M^\delta$, the values of $\mathbf{W}$ are assigned independently, and the values of all other variables $(\mathbf{V} \setminus \mathbf{W})$ only depend on their parents, maintaining all the CSI relations. As a result, $M^\delta$ is compatible with $\mathcal{G}^{\mathcal{L}}$. Also, by construction,

$$
\begin{aligned}
P^{M^\delta}(\mathbf{O} \setminus \mathbf{W}|\mathbf{W} = \mathbf{w}_0) &= P^M(\mathbf{O} \setminus \mathbf{W}), \\
P^{M^\delta}(\mathbf{O} \setminus \mathbf{W}|do(x), \mathbf{W} = \mathbf{w}_0) &= P^M(\mathbf{O} \setminus \mathbf{W}|do(x)).
\end{aligned}
\tag{9}
$$

Next, we write

$$
\begin{aligned}
P^{M^\delta}(y) &= \sum_{\mathbf{w} \in \mathcal{D}_{\mathbf{W}}} P^{M^\delta}(y|\mathbf{w}) P^{M^\delta}(\mathbf{w}) \\
&= P^{M^\delta}(y|\mathbf{W} = \mathbf{w}_0) P^{M^\delta}(\mathbf{W} = \mathbf{w}_0) \\
&+ P^{M^\delta}(y|\mathbf{W} \neq \mathbf{w}_0) P^{M^\delta}(\mathbf{W} \neq \mathbf{w}_0) \\
&= (1 - \delta) P^{M^\delta}(y|\mathbf{W} = \mathbf{w}_0) + \delta P^{M^\delta}(y|\mathbf{W} \neq \mathbf{w}_0).
\end{aligned}
$$

The second term of the right-hand side above is a positive number not larger than $\delta$. Moreover, by construction of $M^\delta$, we have $P^{M^\delta}(y|\mathbf{W} = \mathbf{w}_0) = P^M(y)$. Therefore, we get

$$
(1 - \delta) P^M(y) \leq P^{M^\delta}(y) \leq (1 - \delta) P^M(y) + \delta.
\tag{10}
$$

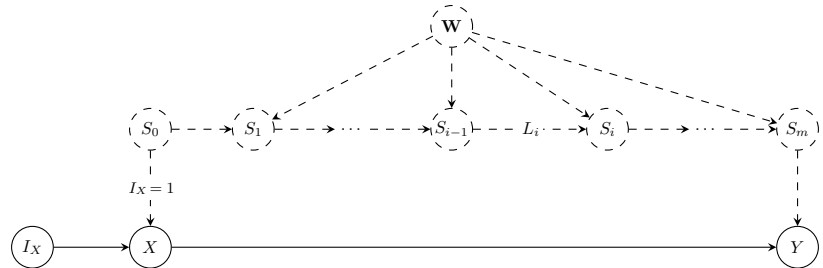

Figure 5: Reduction from 3-SAT to imitability.

On the other hand, for an arbitrary policy $\pi \in \Pi$, depending on the values of $\mathbf{Z} \subseteq \mathbf{Pa}^{\Pi}$,

$$
\begin{aligned}
P^{M^\delta}(y|do(\pi)) &= P^{M^\delta}(y|do(\pi), \mathbf{W} = \mathbf{w}_0)P^{M^\delta}(\mathbf{W} = \mathbf{w}_0|do(\pi)) \\
&\quad + P^{M^\delta}(y|do(\pi), \mathbf{W} \neq \mathbf{w}_0)P^{M^\delta}(\mathbf{W} \neq \mathbf{w}_0|do(\pi)) \\
&= P^{M^\delta}(y|do(\pi), \mathbf{W} = \mathbf{w}_0)P^M(\mathbf{W} = \mathbf{w}_0) \\
&\quad + P^{M^\delta}(y|do(\pi), \mathbf{W} \neq \mathbf{w}_0)P^M(\mathbf{W} \neq \mathbf{w}_0) \\
&= \sum_{x,\mathbf{z}} P^{M^\delta}(y|do(x), \mathbf{z}, \mathbf{W} = \mathbf{w}_0)\pi(x|\mathbf{z}, \mathbf{w}_0)P^{M^\delta}(\mathbf{z}|\mathbf{W} = \mathbf{w}_0)P^M(\mathbf{W} = \mathbf{w}_0) \\
&\quad + \delta P^{M^\delta}(y|do(\pi), \mathbf{W} \neq \mathbf{w}_0) \\
&= \sum_{x,\mathbf{z}} P^M(y|do(x), \mathbf{z}, \mathbf{W} = \mathbf{w}_0)\pi(x|\mathbf{z}, \mathbf{w}_0)P^M(\mathbf{z}|\mathbf{W} = \mathbf{w}_0)P^M(\mathbf{W} = \mathbf{w}_0) \\
&\quad + \delta P^{M^\delta}(y|do(\pi), \mathbf{W} \neq \mathbf{w}_0) \\
&= P^M(y|do(\pi), \mathbf{W} = \mathbf{w}_0)P^M(\mathbf{W} = \mathbf{w}_0) + \delta P^{M^\delta}(y|do(\pi), \mathbf{W} \neq \mathbf{w}_0) \\
&= P^M(y|do(\pi), \mathbf{W} = \mathbf{w}_0)P^M(\mathbf{W} = \mathbf{w}_0) + P^M(y|do(\pi), \mathbf{W} \neq \mathbf{w}_0)P^M(\mathbf{W} \neq \mathbf{w}_0) \\
&\quad - P^M(y|do(\pi), \mathbf{W} \neq \mathbf{w}_0)P^M(\mathbf{W} \neq \mathbf{w}_0) + \delta P^{M^\delta}(y|do(\pi), \mathbf{W} \neq \mathbf{w}_0) \\
&= P^M(y|do(\pi)) - \delta(P^M(y|do(\pi), \mathbf{W} \neq \mathbf{w}_0) - P^{M^\delta}(y|do(\pi), \mathbf{W} \neq \mathbf{w}_0)),
\end{aligned}
$$

and as a result,

$$
P^M(y|do(\pi)) - \delta \leq P^{M^\delta}(y|do(\pi)) \leq P^M(y|do(\pi)) + \delta. \tag{11}
$$

Combining Equations (10) and (11), we get

$$
\begin{aligned}
\left(P^M(y) - P^M(y|do(\pi))\right) - \delta P^M(y) - \delta & \\
\leq P^{M^\delta}(y) - P^{M^\delta}(y|do(\pi)) &\leq \\
\left(P^M(y) - P^M(y|do(\pi))\right) - \delta P^M(y) + \delta,
\end{aligned}
$$

and consequently,

$$
\left(P^M(y) - P^M(y|do(\pi))\right) - 2\delta \leq P^{M^\delta}(y) - P^{M^\delta}(y|do(\pi)) \leq \left(P^M(y) - P^M(y|do(\pi))\right) + 2\delta,
$$

which combined with Equation (8) results in the following inequality:

$$
|P^{M^\delta}(y) - P^{M^\delta}(y|do(\pi))| \geq \epsilon - 2\delta = \frac{\epsilon}{2} > 0.
$$

To sum up, we showed that there exists model $M^\delta \in \mathcal{M}_{\langle \mathcal{G}^{\mathcal{L}} \rangle}$, such that for every $\pi \in \Pi$, $P^{M^\delta}(y|do(\pi)) \neq P^{M^\delta}(y)$ which is in contradiction with the imitability assumption. Hence, $P(y)$ is imitable w.r.t. $\langle \mathcal{G}^{\mathcal{L}_{\mathbf{w}}}_{\mathbf{w}}, \Pi \rangle$ for every $\mathbf{w} \in \mathcal{D}_{\mathbf{W}}$ of any arbitrary subset of variables $\mathbf{W} \subseteq \mathbf{V} \setminus Y$. $\quad\square$

**Theorem 3.8.** *Given an LDAG $\mathcal{G}^{\mathcal{L}}$ and a policy space $\Pi$, deciding the imitability of $P(y)$ w.r.t. $\langle \mathcal{G}^{\mathcal{L}}, \Pi \rangle$ is NP-hard.*

*Proof.* Our proof exploits ideas similar to the proof of hardness of causal identification under CSIs by [39]. We show a polynomial-time reduction from the 3-SAT problem, which is a well-known NP-hard problem [20], to the imitability problem. Let an instance of 3-SAT be defined as follows. A formula $F$ in the conjunctive normal form with clauses $S_1, \ldots, S_m$ is given, where each clause $S_i$ has 3 literals from the set of binary variables $W_1, \ldots, W_k$ and their negations, $\neg W_1, \ldots, \neg W_k$. The decision problem of 3-SAT is to determine whether there is a realization of the binary variables $W_1, \ldots, W_k$ that *satisfies* the given formula $F$, i.e., makes $F$ true. We reduce this problem to an instance of imitability problem with CSIs as follows. Define an LDAG with one vertex corresponding to each clause $S_i$ for $1 \leq i \leq m$, one vertex $\mathbf{W}$ which corresponds to the set of binary variables $\{W_1, \ldots, W_k\}$, and three auxiliary vertices $S_0, X$, and $Y$. Vertex $\mathbf{W}$ is a parent of every $S_i$ for $1 \leq i \leq m$. There is an edge from $S_{i-1}$ to $S_i$ for $1 \leq i \leq m$. Also, $S_0$ is a parent of $X$, and $X$ and $S_m$ are parents of $Y$. As for the labels, the label on each edge $S_{i-1} \rightarrow S_i$ is that this edge is absent if the assignment of $\mathbf{W}$ does not satisfy the clause $S_i$. Constructing this LDAG is clearly polynomial-time, as it only requires time in the order of the number of variables $\{W_1, \ldots, W_k\}$ and the number of clauses $\{S_1, \ldots, S_m\}$. We claim that $P(y)$ is imitable in this LDAG if and only if the 3-SAT instance is unsatisfiable.

**Only if part.** Suppose that the 3-SAT instance is satisfiable. There exists a realization $\mathbf{w}^*$ of the binary variables $\{W_1, \ldots, W_k\}$ that satisfies the formula $F$. Define a model $M$ over the variables of the LDAG as follows. $\mathbf{W}$ has the value $\mathbf{w}^*$ with probability 0.6 ($P^M(\mathbf{W} = \mathbf{w}^*) = 0.6$), and the rest of the probability mass is uniformly distributed over other realizations of $\mathbf{W}$ (i.e., each with probability $\frac{0.4}{2^k - 1}$). $S_0$ is a Bernoulli random variable with parameter $\frac{1}{2}$. $S_i$ for $1 \leq i \leq m$ is equal to $S_{i-1}$ if the edge $S_{i-1} \rightarrow S_i$ is present (the clause $S_i$ is satisfied by the realization of $\mathbf{W}$), and $S_i = 0$ otherwise. $X$ is equal to $S_0$, and $Y$ is defined as $Y = \neg X \oplus S_m$. Note that model $M$ is compatible with the LDAG defined above. We claim that under this model, for every policy $\pi \in \Pi$, $P^M(y|do(\pi)) \neq P^M(y)$. That is, $P(y)$ is not imitable.

Let $\mathcal{D}_{\mathbf{W}}^s$ and $\mathcal{D}_{\mathbf{W}}^u$ be the partitioning of the domain of $\mathbf{W}$ into two disjoint parts which satisfy and do not satisfy the formula $F$, respectively. Note that $\mathbf{w}^* \in \mathcal{D}_{\mathbf{W}}^s$. For any realization $\mathbf{w} \in \mathcal{D}_{\mathbf{W}}^s$, observed values of $Y$ in $M$ is always equal to 1 (because $Y = \neg S_0 \oplus S_0$), i.e., $P^M(Y = 1|\mathbf{w}) = 1$. On the other hand, for any realization $\mathbf{w} \in \mathcal{D}_{\mathbf{W}}^u$, there exists a clause $S_i$ which is not satisfied, and therefore $S_m = 0$, and $Y = \neg X$. Therefore, $P^M(Y = 1|\mathbf{w}) = P^M(X = 0) = \frac{1}{2}$. From the total probability law,

$$
\begin{aligned}
P^M(Y = 1) &= \sum_{\mathbf{w} \in \mathcal{D}_{\mathbf{W}}^s} P(Y = 1|\mathbf{w})P(\mathbf{w}) + \sum_{\mathbf{w} \in \mathcal{D}_{\mathbf{W}}^u} P(Y = 1|\mathbf{w})P(\mathbf{w}) \\
&= \sum_{\mathbf{w} \in \mathcal{D}_{\mathbf{W}}^s} P(\mathbf{w}) + \frac{1}{2} \sum_{\mathbf{w} \in \mathcal{D}_{\mathbf{W}}^u} P(\mathbf{w}) \\
&= \frac{1}{2}\Big( \sum_{\mathbf{w} \in \mathcal{D}_{\mathbf{W}}^s} P(\mathbf{w}) + \sum_{\mathbf{w} \in \mathcal{D}_{\mathbf{W}}^u} P(\mathbf{w}) \Big) + \frac{1}{2} \sum_{\mathbf{w} \in \mathcal{D}_{\mathbf{W}}^s} P(\mathbf{w}) \\
&= \frac{1}{2} + \frac{1}{2} \sum_{\mathbf{w} \in \mathcal{D}_{\mathbf{W}}^s} P(\mathbf{w}) \geq \frac{1}{2} + \frac{1}{2}P(\mathbf{w}^*) = 0.8,
\end{aligned}
\tag{12}
$$

where we dropped the superscript $M$ for better readability. Now consider an arbitrary policy $\pi \in \Pi$. For any realization $\mathbf{w} \in \mathcal{D}_{\mathbf{W}}^s$, $Y = \neg X \oplus S_0$, where $S_0$ is a Bernoulli variable with parameter $\frac{1}{2}$ independent of $X$. As a result, $P^M(Y = 1|do(\pi), \mathbf{w}) = P^M(S = X) = \frac{1}{2}$. On the other hand, for any realization $\mathbf{w} \in \mathcal{D}_{\mathbf{W}}^u$, there exists a clause $S_i$ which is not satisfied, and therefore $S_m = 0$, and $Y = \neg X$. Therefore, $P^M(Y = 1|do(\pi), \mathbf{w}) = P^M(X = 0|do(\pi)) = \pi(X = 0)$. Using the total probability law again, and noting that $P^M(\mathbf{w}|do(\pi)) = P^M(\mathbf{w})$,

$$
\begin{aligned}
P^M(Y = 1|do(\pi)) &= \sum_{\mathbf{w} \in \mathcal{D}_{\mathbf{W}}^s} P(Y = 1|do(\pi), \mathbf{w})P(\mathbf{w}|do(\pi)) \\
&\quad + \sum_{\mathbf{w} \in \mathcal{D}_{\mathbf{W}}^u} P(Y = 1|do(\pi), \mathbf{w})P(\mathbf{w}|do(\pi)) \\
&= \frac{1}{2} \sum_{\mathbf{w} \in \mathcal{D}_{\mathbf{W}}^s} P(\mathbf{w}) + \pi(X = 0) \sum_{\mathbf{w} \in \mathcal{D}_{\mathbf{W}}^u} P(\mathbf{w}),
\end{aligned}
$$

where we dropped the superscript $M$ for better readability. Now define $q = \sum_{\mathbf{w} \in \mathcal{D}_\mathbf{W}^s} P^M(\mathbf{w}) = 1 - \sum_{\mathbf{w} \in \mathcal{D}_\mathbf{W}^u} P(\mathbf{w})$, where we know $q \geq 0.6$ since $\mathbf{w}^* \in \mathcal{D}_\mathbf{W}^s$. From the equation above we have

$$P^M(Y = 1|do(\pi)) = \frac{1}{2}q + \pi(X = 0)(1 - q), \tag{13}$$

where $0.6 \leq q \leq 1$ and $0 \leq \pi(X = 0) \leq 1$. We claim that $P(Y = 1|do(\pi)) < 0.7$. Consider two cases: if $\pi(X = 0) \leq \frac{1}{2}$, then from Equation (13),

$$P(Y = 1|do(\pi)) \leq \frac{1}{2}q + \frac{1}{2}(1 - q) = 0.5 < 0.7.$$

Otherwise, assume $\pi(X = 0) > \frac{1}{2}$. Rewriting Equation (13),

$$P(Y = 1|do(\pi)) = q(\frac{1}{2} - \pi(X = 0)) + \pi(X = 0) < 0.6(\frac{1}{2} - \pi(X = 0)) + \pi(X = 0)$$

$$= 0.3 + 0.4\pi(X = 0) \leq 0.7.$$

We showed that for any arbitrary policy $\pi \in \Pi$,

$$P^M(Y = 1|do(\pi)) < 0.7. \tag{14}$$

Comparing this to Equaiton (12) completes the proof.

**If part.** Let $I_X$ be the intervention vertex corresponding to $X$, i.e., $I_X = 0$ and $I_X = 1$ indicate that $X$ is passively observed (determined by its parents) and actively intervened upon (independent of its parents), respectively (refer to Figure 5). Assume that the 3-SAT instance is unsatisfiable. We claim that $\pi(x) = P(x)$ is an imitating policy for $P(y)$. Since the 3-SAT is unsatisfiable, for any context $\mathbf{w} \in \mathcal{D}_\mathbf{W}$, there exists $1 \leq i \leq m$ such that the edge $S_{i-1} \to S_i$ is absent. Then, we have $I_X \perp Y|X, \mathbf{W} = \mathbf{w}$ in $\mathcal{G}$ for every $\mathbf{w} \in \mathcal{D}_\mathbf{W}$, then $I_X \perp Y|X, \mathbf{W}$ in $\mathcal{G}$. Moreover, since the d-separation $I_X \perp_\mathcal{G} \mathbf{W}$ holds, by contraction, we have $I_X \perp Y, \mathbf{W}|X$ in $\mathcal{G}$. As a result, $I_X \perp Y|X$ in $\mathcal{G}$. Therefore,

$$P(y|do(\pi)) = \sum_{x \in \mathcal{D}_X} P(y|do(x))\pi(x) = \sum_{x \in \mathcal{D}_X} P(y|x, I_X = 1)P(x) = \sum_{x \in \mathcal{D}_X} P(y|x, I_X = 0)P(x)$$

$$= \sum_{x \in \mathcal{D}_X} P(y|x)P(x) = \sum_{x \in \mathcal{D}_X} P(y|x)P(x) = P(y).$$

$\square$

**Proposition 3.9.** *Given an LDAG $\mathcal{G}^\mathcal{L}$ where $\mathbf{Pa}(\mathbf{C}(\mathcal{L})) \subseteq \mathbf{C}(\mathcal{L})$ and a policy space $\Pi$, let $Y$ be an arbitrary variable in $\mathcal{G}^\mathcal{L}$. $P(y)$ is imitable w.r.t. $\langle \mathcal{G}^\mathcal{L}, \Pi \rangle$ if and only if $P(y)$ is imitable w.r.t. $\langle \mathcal{G}_\mathbf{c}, \Pi \rangle$, for every $\mathbf{c} \in \mathcal{D}_{\mathbf{C}(\mathcal{L})}$.*

*Proof.* It suffices to show that for an arbitrary $M \in \mathcal{M}_{\langle \mathcal{G}^\mathcal{L} \rangle}$, there exists a policy $\pi \in \Pi$ uniquely computable from $P(\mathbf{O})$ such that $P^M(y|do(\pi)) = P^M(y)$. For ease of notation, let $\mathbf{C} := \mathbf{C}(\mathcal{L})$. For every $\mathbf{c} \in \mathcal{D}_\mathbf{C}$, construct an SCM $M_\mathbf{c}$ over $\mathbf{V}$ by setting $\mathbf{C} = \mathbf{c}$, and replacing every occurrence of variables $\mathbf{C}$ by the constant value $\mathbf{c}$ in the equations of $M$. Therefore, $M_\mathbf{c}$ is compatible with $\mathcal{G}_\mathbf{c}$ and

$$P^{M_\mathbf{c}}(\mathbf{O}) = P^M(\mathbf{O}|do(\mathbf{c})).$$

By assumption, we know that for every $\mathbf{c} \in \mathcal{D}_\mathbf{C}$, there exists a policy $\pi_\mathbf{c} \in \Pi$ uniquely computable from $P(\mathbf{O})$ such that

$$P^{M_\mathbf{c}}(y|do(\pi_\mathbf{c})) = P^{M_\mathbf{c}}(y). \tag{15}$$

Suppose $\pi_c$ relies on the values of $\mathbf{Z}_\mathbf{c} \subseteq \mathbf{Pa}^\Pi$. We can write

$$P^{M_\mathbf{c}}(y|do(\pi_\mathbf{c})) = P^M(y|do(\pi_\mathbf{c}), do(\mathbf{c}))$$

$$= \sum_{x \in \mathcal{D}_X, \mathbf{z}_\mathbf{c} \in \mathcal{D}_{\mathbf{Z}_\mathbf{c}}} P^M(y|do(x), do(\mathbf{c}), \mathbf{z}_\mathbf{c})P^M(x|do(\pi_\mathbf{c}), do(\mathbf{c}), \mathbf{z}_\mathbf{c})P^M(\mathbf{z}_\mathbf{c}|do(\mathbf{c}))$$

$$\overset{(a)}{=} \sum_{x \in \mathcal{D}_X, \mathbf{z}_\mathbf{c} \in \mathcal{D}_{\mathbf{Z}_\mathbf{c}}} P^M(y|do(x), \mathbf{c}, \mathbf{z}_\mathbf{c})P^M(x|do(\pi_\mathbf{c}), \mathbf{c}, \mathbf{z}_\mathbf{c})P^M(\mathbf{z}_\mathbf{c}|\mathbf{c})$$

$$= P^M(y|do(\pi_\mathbf{c}), \mathbf{c}), \tag{16}$$

where the first and third parts of (a) hold since $Y \perp \mathbf{C}|X, \mathbf{Z}_{\mathbf{c}}$ in $\mathcal{G}_{\overline{X}\underline{\mathbf{C}}}$ and $\mathbf{Pa}(\mathbf{C}) \subseteq \mathbf{C}$. For the second part, let $M_{\pi_{\mathbf{c}}}$ be the model obtained from substituting the model of $X$ given its parents by $\pi_{\mathbf{c}}$ in the equations of $M$. Then, we get

$$P^M(x|do(\pi_{\mathbf{c}}), do(\mathbf{c}), \mathbf{z}_{\mathbf{c}}) = P^{M_{\pi_{\mathbf{c}}}}(x|do(\mathbf{c}), \mathbf{z}_{\mathbf{c}}) = P^{M_{\pi_{\mathbf{c}}}}(x|\mathbf{c}, \mathbf{z}_{\mathbf{c}}) = P^M(x|do(\pi_{\mathbf{c}}), \mathbf{c}, \mathbf{z}_{\mathbf{c}}).$$

Note that, since $\mathbf{Pa}(\mathbf{C}) \subseteq \mathbf{C}$, $P^{M_{\pi_{\mathbf{c}}}}(x|do(\mathbf{c}), \mathbf{z}_{\mathbf{c}}) = P^{M_{\pi_{\mathbf{c}}}}(x|\mathbf{c}, \mathbf{z}_{\mathbf{c}})$.

Furthermore,

$$P^{M_{\mathbf{c}}}(y) = P^M(y|do(\mathbf{c})) = P^M(y|\mathbf{c}). \tag{17}$$

Combining Equations (16), (17) and (15) we get the following for every $\mathbf{c} \in \mathcal{D}_{\mathbf{C}}$

$$P^M(y|do(\pi_{\mathbf{c}}), \mathbf{c}) = P^M(y|\mathbf{c}). \tag{18}$$

Next, define a policy $\pi^*(x|\mathbf{Pa}^{\Pi}) := \sum_{\mathbf{c} \in \mathcal{D}_{\mathbf{C}}} \mathbb{1}_{\{(\mathbf{Pa})_{\mathbf{C}}^{\Pi} = \mathbf{c}\}} \pi_{\mathbf{c}}$, where $\mathbb{1}_{\{(\mathbf{Pa})_{\mathbf{C}}^{\Pi} = \mathbf{c}\}}$ is an indicator function such that $\mathbb{1}_{\mathbf{c}} = 1$ when $\mathbf{C} = \mathbf{c}$, and is equal to zero otherwise. Now, we can write

$$P^M(y|do(\pi^*))$$
$$= \sum_{\mathbf{c} \in \mathcal{D}_{\mathbf{C}}} P^M(y|do(\pi^*), \mathbf{c}) P^M(\mathbf{c}|do(\pi^*))$$
$$= \sum_{\mathbf{c} \in \mathcal{D}_{\mathbf{C}}} P^M(y|do(\pi_{\mathbf{c}}), \mathbf{c}) P^M(\mathbf{c})$$
$$= \sum_{\mathbf{c} \in \mathcal{D}_{\mathbf{C}}} P^M(y|\mathbf{c}) P^M(\mathbf{c}) = P^M(y),$$

where the last line holds due to Equation (18) and the fact that $\mathbf{Pa}(\mathbf{C}) \subseteq \mathbf{C}$. We proved that there exists a policy $\pi^* \in \Pi$ uniquely computable from $P(\mathbf{O})$ such that $P^M(y|do(\pi^*)) = P^M(y)$. Hence, $P(y)$ is imitable w.r.t. $\langle \mathcal{G}^{\mathcal{L}}, \Pi, \rangle$. $\qquad \square$

**Proposition 4.2.** *Given an LDAG $\mathcal{G}^{\mathcal{L}}$ and a policy space $\Pi$, let $\mathbf{C}$ be a subset of $\mathbf{Pa}^{\Pi} \cap \mathbf{C}(\mathcal{L})$. If for every realization $\mathbf{c} \in \mathcal{D}_{\mathbf{C}}$ at least one of the following holds, then $P(y)$ is imitable w.r.t. $\langle \mathcal{G}^{\mathcal{L}}, \Pi, P(\mathbf{O}) \rangle$.*

- *There exists a subset $\mathbf{S}_{\mathbf{c}}$ such that $X \perp Y|\{\mathbf{S}_{\mathbf{c}}, \mathbf{C}\}$ in $\mathcal{G}^{\mathcal{L}}(\mathbf{c}) \cup \Pi$, and $P(\mathbf{S}_{\mathbf{c}}|do(\pi), \mathbf{C} = \mathbf{c}) = P(\mathbf{S}_{\mathbf{c}}|\mathbf{C} = \mathbf{c})$ has a solution $\pi_{\mathbf{c}} \in \Pi$.*

- *There exists $\mathbf{Z}_c \subseteq \mathbf{Pa}^{\Pi}$ such that $X \perp Y|\{\mathbf{Z}_{\mathbf{c}}, \mathbf{C}\}$ in $\mathcal{G}^{\mathcal{L}}(\mathbf{c})_{\underline{X}}$.*

*Proof.* Let $M \in \mathcal{M}_{\langle \mathcal{G}^{\mathcal{L}} \rangle}$ and $P^M(\mathbf{O}) = P(\mathbf{O})$, i.e., $M$ induces the observational distribution $P(\mathbf{O})$, be an arbitrary model. It suffices to show that there is a policy $\pi \in \Pi$ uniquely computable from $P(\mathbf{O})$ such that $P^M(y|do(\pi)) = P^M(y)$. To do so, first, partition $\mathcal{D}_{\mathbf{C}}$ into two subsets $\mathcal{D}_{\mathbf{C}}^1$ and $\mathcal{D}_{\mathbf{C}}^2$ such that $\mathcal{D}_{\mathbf{C}} = \mathcal{D}_{\mathbf{C}}^1 \cup \mathcal{D}_{\mathbf{C}}^2$ and for every $\mathbf{c} \in \mathcal{D}_{\mathbf{C}}^1$ the first condition of the lemma holds, whereas for every $\mathbf{c} \in \mathcal{D}_{\mathbf{C}}^2$ the second condition does.

For each $\mathbf{c} \in \mathcal{D}_{\mathbf{C}}^1$, construct model $M_{\pi_{\mathbf{c}}}$ by substituting the model of $X$ given its parents by $\pi_{\mathbf{c}}$ in $M$. Now, we get the following for each $\mathbf{c} \in \mathbf{D}_{\mathbf{C}}^1$,

$$P^M(y|do(\pi_{\mathbf{c}}), \mathbf{c}) = P^{M_{\pi_{\mathbf{c}}}}(y|\mathbf{c}) = \sum_{\mathbf{s}_{\mathbf{c}} \in \mathcal{D}_{\mathbf{S}_{\mathbf{c}}}} P^{M_{\pi_{\mathbf{c}}}}(y|\mathbf{c}, \mathbf{s}_{\mathbf{c}}) P^{M_{\pi_{\mathbf{c}}}}(\mathbf{s}_{\mathbf{c}}|\mathbf{c})$$

$$\stackrel{(a)}{=} \sum_{\mathbf{s}_{\mathbf{c}} \in \mathcal{D}_{\mathbf{S}_{\mathbf{c}}}} P^{M_{\pi_{\mathbf{c}}}}(y|do(x), \mathbf{c}, \mathbf{s}_{\mathbf{c}}) P^{M_{\pi_{\mathbf{c}}}}(\mathbf{s}_{\mathbf{c}}|\mathbf{c})$$

$$= \sum_{\mathbf{s}_{\mathbf{c}} \in \mathcal{D}_{\mathbf{S}_{\mathbf{c}}}} P^M(y|do(x), \mathbf{c}, \mathbf{s}_{\mathbf{c}}) P^{M_{\pi_{\mathbf{c}}}}(\mathbf{s}_{\mathbf{c}}|\mathbf{c}) \tag{19}$$

$$\stackrel{(b)}{=} \sum_{\mathbf{s}_{\mathbf{c}} \in \mathcal{D}_{\mathbf{S}_{\mathbf{c}}}} P^M(y|\mathbf{c}, \mathbf{s}_{\mathbf{c}}) P^{M_{\pi_{\mathbf{c}}}}(\mathbf{s}_{\mathbf{c}}|\mathbf{c})$$

$$\stackrel{(c)}{=} \sum_{\mathbf{s}_{\mathbf{c}} \in \mathcal{D}_{\mathbf{S}_{\mathbf{c}}}} P^M(y|\mathbf{c}, \mathbf{s}_{\mathbf{c}}) P^M(\mathbf{s}_{\mathbf{c}}|\mathbf{c}) = P^M(y|\mathbf{c}).$$

---
**Algorithm 3** Find Possible $\pi$-backdoor admissible set
---
1: **function** FINDSEP $(\mathcal{G}, \Pi, X, Y, \mathbf{C})$
2:     **if C** is not specified **then**
3:         $\mathbf{C} \leftarrow \emptyset$
4:     Set $\mathbf{Z} = \mathbf{An}(\{X, Y\} \cup \mathbf{C}) \cap (\mathbf{Pa}^{\Pi})$
5:     **if TESTSEP** $(\mathcal{G}_{\underline{X}}, X, Y, \mathbf{Z})$ **then**
6:         **return Z**
7:     **else**
8:         **return** FAIL
---

where (a) and (b) hold since $Y \perp X|\mathbf{S_c}, \mathbf{C}$ in $\mathcal{G}^{\mathcal{L}}(\mathbf{c}) \cup \Pi$ and in $\mathcal{G}^{\mathcal{L}}(\mathbf{c})$, respectively (or $Y \perp\!\!\!\perp X|\mathbf{S_c}, \mathbf{c}$ in $M^{\pi_\mathbf{c}}$ and $M$). Moreover, (c) holds because $P^M(\mathbf{S_c}|do(\pi_\mathbf{c}), \mathbf{c}) = P^M(\mathbf{S_c}|\mathbf{c})$.

Next, for each $\mathbf{c} \in \mathcal{D}_\mathbf{C}^2$, let $\pi_\mathbf{c}(x|\mathbf{Pa}^{\Pi}) = P^M(x|\mathbf{Z_c}, \mathbf{c})$, then we get

$$P^M(y|do(\pi_\mathbf{c}), \mathbf{c}) = \sum_{x \in \mathcal{D}_X, \mathbf{z_c} \in \mathcal{D}_{\mathbf{Z_c}}} P^M(y|do(x), \mathbf{z_c}, \mathbf{c}) P^M(x|\mathbf{z_c}, \mathbf{c}) P^M(\mathbf{z_c}|\mathbf{c})$$

$$= \sum_{x \in \mathcal{D}_X, \mathbf{z_c} \in \mathcal{D}_{\mathbf{Z_c}}} P^M(y|x, \mathbf{z_c}, \mathbf{c}) P^M(x|\mathbf{z_c}, \mathbf{c}) P^M(\mathbf{z_c}|\mathbf{c}) = P^M(y|\mathbf{c}),$$

where the second equation holds since $X \perp Y|\mathbf{Z_c}, \mathbf{C}$ in $\mathcal{G}^{\mathcal{L}}(\mathbf{c})_{\underline{X}}$.

Now, define a policy $\pi^*(x|\mathbf{Pa}^{\Pi}) := \sum_{\mathbf{c} \in \mathcal{D}_\mathbf{C}} \mathbb{1}_{\{(\mathbf{Pa})_\mathbf{C}^{\Pi} = \mathbf{c}\}} \pi_\mathbf{c}$.

We get the following where the second line follows from Equation (19) and the fact that $\mathbf{C} \cap \mathbf{De}(X) = \emptyset$.

$$P^M(y|do(\pi^*)) = \sum_{\mathbf{c} \in \mathcal{D}_\mathbf{C}} P^M(y|do(\pi^*), \mathbf{c}) P^M(\mathbf{c}|do(\pi^*))$$

$$= \sum_{\mathbf{c} \in \mathcal{D}_\mathbf{C}} P^M(y|do(\pi_\mathbf{c}), \mathbf{c}) P^M(\mathbf{c})$$

$$= \sum_{\mathbf{c} \in \mathcal{D}_\mathbf{C}} P^M(y|\mathbf{c}) P^M(\mathbf{c}) = P^M(y).$$

The above equation implies that there is a policy $\pi^* \in \Pi$ such that $P^M(y|do(\pi^*)) = P^M(y)$ where $M$ is an arbitrary model compatible with $\mathcal{G}^{\mathcal{L}}$. Therefore, $P(y)$ is imitable w.r.t. $\langle \mathcal{G}^{\mathcal{L}}, \Pi \rangle$ and $\pi^*$ is an imitating policy for that.

$\square$

## B   FindSep Algorithm

*FindSep* was used as a subroutine in Alg. 1. This function is summarized here as Algorithm 3, which takes a DAG $\mathcal{G}$, variables $X$ and $Y$, and a subset of variables $\mathbf{C}$ as inputs. It finds a separating set containing $\mathbf{C}$, if exists, in $\mathcal{G}_{\underline{X}}$ between $X$ and $Y$. To do so, it constructs $\mathbf{Z}$ in line 4. According to lemma 3.4 of [40] where $I = \mathbf{C}$ and $R = \mathbf{Pa}^{\Pi}$, if such separating set exists, $\mathbf{Z}$ is a separating set.

## C   Example for Algorithm 2

We present a simple counterexample to illustrate that Algorithm 2 is not complete, in the sense that it may happen that an instance is imitable w.r.t. $\langle \mathcal{G}^{\mathcal{L}}, \Pi, P(\mathbf{O}) \rangle$, but Algorithm 2 fails to find an imitating policy. To that end, take the instrumental variable graph depicted in Figure 6. It is evident that the set of equations

$$P(y|do(\pi)) = P(y|X = 0, C = 0)\pi(X = 0) + P(y|X = 1, C = 0)\pi(X = 1)$$
$$= P(y|X = 0)P(X = 0) + P(y|X = 1)P(X = 1) = P(y)$$

for both values of $y = 0$ and $y = 1$ is a set of linear equations with two equations in two parameters, namely $\pi(X = 0)$ and $\pi(X = 1)$, which always has a solution. This indeed indicates that this

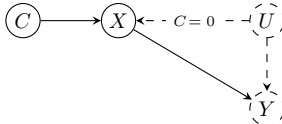

Figure 6: An example where Algorithm 2 fails to identify an imitating policy, despite the fact that there exists one such policy.

instance is imitable, i.e., there always exists at least one imitation policy. However, Algorithm 2 would not be able to find any such policy.

## D   Experimental setup

For the experiments of Section 5.2, we worked with an SCM in which

$$
\begin{aligned}
&C \sim Be(0.05), \\
&P(Y = 0|C = 1) = 0.2, \\
&P(Y = 1|C = 1) = 0.5, \\
&P(Y = 2|C = 1) = 0.3, \\
&T \sim Be(0.4), \\
&U_1 \sim Be(0.8), \\
&X|T = 0, U_1 = 0 \sim Be(0.7), \\
&X|T = 0, U_1 = 1 \sim Be(0.7), \\
&X|T = 1, U_1 = 0 \sim Be(0), \text{ and,} \\
&X|T = 1, U_1 = 1 \sim Be(1).
\end{aligned}
$$

Moreover, we had

$$
\begin{aligned}
&S|C = 0, X = 0, U_1 = 0 \sim Be(1), \\
&S|C = 0, X = 0, U_1 = 1 \sim Be(0), \\
&S|C = 0, X = 1, U_1 = 0 \sim Be(0), \\
&S|C = 0, X = 1, U_1 = 1 \sim Be(1).
\end{aligned}
$$

And finally,

$$
\begin{aligned}
&P(Y = 0|C = 0, S = 0) = 0.8, \\
&P(Y = 1|C = 0, S = 0) = 0.1, \\
&P(Y = 2|C = 0, S = 0) = 0.1, \\
&P(Y = 0|C = 0, S = 1) = 0.05, \\
&P(Y = 1|C = 0, S = 1) = 0.2, \text{ and,} \\
&P(Y = 2|C = 0, S = 1) = 0.75.
\end{aligned}
$$

