# OpenReview forum: "Causal Imitability Under Context-Specific Independence Relations"
_NeurIPS.cc/2023/Conference — NeurIPS 2023 poster_

### Official Review · Reviewer_puDh · 2023-07-03

**Soundness:** 3 good
**Presentation:** 2 fair
**Contribution:** 2 fair
**Rating:** 6
**Confidence:** 3

**Summary:**

The paper studies causal imitation learning. In particular, it extends traditional causal graphs with context specific independence. Although the original causal imitation learning problem can be reduced to d-seperation test, causal imitation learning with context specific independence is NP hard. But under other conditions, causal imitation learning with CSI can be solved more efficiently.

**Strengths:**

The problem is well motivated and theoretical contributions seem solid.

**Weaknesses:**

1. Theoretical results rely heavily on (Zhang [et.al](http://et.al), 2020) and might appear derivative. It would be better if the authors can explain a bit more on the contributions in contrast to existing works, especially in terms of identifiability, because NP-hardness is not surprising.
2. Experiments are relatively weak as they are only tested on synthetic datasets.

**Questions:**


1. Context-specific independence seems like a specific form of structural equations? For instance, in the wage example, CSI can be incorporated into how the function w = f(e, u) is defined. I guess it does not work with Pearl’s do-calculus because of a violation of faithfulness? My point is that it does not seem like causal graphs need refinement to incorporate CSI?
2. In line 134, replacing fx with stochastic mapping \pi in mentioned as soft interventions. But as far as I know, soft interventions do not allow adding edges between intervened variable and its non-parent?
3. In definition 3.5, what’s intuition that the edges incident to W are deleted?

**Limitations:**

Not applicable.

---

> ### Author Rebuttal · Authors · 2023-08-07
>
> We thank the reviewer for their valuable feedback, and we are delighted to learn that they found our contribution to be solid. Below, the main points and questions raised by the reviewer are addressed.
>
> ---
> >Experiments are relatively weak as they are only tested on synthetic datasets.
>
> Indeed, our experiments were primarily aimed at illustrating the potential benefits of taking context-specific information (CSIs) into account in the imitation learning process, particularly in terms of 'achievability' of imitation learning. We agree that the current experiments are limited to synthetic datasets, and we acknowledge that testing the proposed algorithms on real-world datasets is essential to validate their practical applicability.
> As we move forward, we plan to address this limitation. Developing more efficient algorithms for handling CSIs as well as conducting extensive experiments on real-world datasets to evaluate the performance and efficiency of our approach in practical scenarios will be a part of our future work to enhance the overall usability and scalability of the proposed methods.
>
> ---
> > Theoretical results rely heavily on (Zhang et.al, 2020) and might appear derivative. It would be better if the authors can explain a bit more on the contributions in contrast to existing works, especially in terms of identifiability, because NP-hardness is not surprising.
>
> While we acknowledge that we drew inspiration from previous work, the introduction of CSIs and their influence on imitability are the key contributions that distinguish our approach and open up new possibilities for imitation learning in complex scenarios. In our experiments, particularly illustrated in Figure 4, we demonstrate how accounting for CSIs can significantly increase the number of instances that become imitable, showcasing the practical benefits of our proposed framework. We will emphasize these aspects in the revised version.
>
> ---
> >1. Context-specific independence seems like a specific form of structural equations? For instance, in the wage example, CSI can be incorporated into how the function w = f(e, u) is defined. I guess it does not work with Pearl’s do-calculus because of a violation of faithfulness? My point is that it does not seem like causal graphs need refinement to incorporate CSI?
>
> It is absolutely correct that context-specific independence (CSI) is inherent in the functional form of the structural causal model (SCM). However, when it comes to the graphical representation of causal relationships, traditional causal graphs, such as directed acyclic graphs (DAGs), are not equipped to directly incorporate CSIs without additional refinement.
> CSI relations imply conditional independence relationships between certain subsets of variables given specific contexts. While this information can be encoded in the functional form of the SCM, it is not explicitly captured in the standard graphical representation. Traditional DAGs represent the causal relationships between variables in a global sense, but they do not distinguish between different contexts in which these relationships might vary.
> We took advantage of the concept of labeled DAGs to represent context-specific causal relationships explicitly, making it possible to distinguish between different contexts and their corresponding conditional independencies.
>
> ---
> >2.  In line 134, replacing fx with stochastic mapping $\pi$ in mentioned as soft interventions. But as far as I know, soft interventions do not allow adding edges between intervened variable and its non-parent?
>
> Indeed in the literature, a soft intervention is often used to refer specifically to interventions that do not change the parents of a variable. In our work, we used the term "soft intervention" in a broader sense to encompass any form of intervention that may or may not change the set of parents. We will clarify this terminology in the revision to ensure consistency with the existing literature. We thank the reviewer for bringing this to our attention.
>
> >3. In definition 3.5, what’s intuition that the edges incident to W are deleted?
>
> We aimed at constructing a subgraph such that imitability in the original graph would imply imitability in the subgraph (which, in our work is the context-induced subgraph of definition 3.5).
> Equivalently, any non-imitable model over the subgraph must imply non-imitability in the original graph.
> To have the flexibility in defining a non-imitable model, intuitively speaking, we would like the context variables $W$ to be completely detached from the rest of the model. This allows us to define a structural causal model (SCM) on the subgraph, where the model over $W$ is independent of the model over $V\setminus W$. For further details, please see the proof of Lemma 3.6.

---

> > ### Comment · Reviewer_puDh · 2023-08-17
> >
> > Thanks for your reply! I don't have any further questions at this moment.

---

### Official Review · Reviewer_g52k · 2023-07-05

**Soundness:** 4 excellent
**Presentation:** 3 good
**Contribution:** 3 good
**Rating:** 7
**Confidence:** 3

**Summary:**

This paper studies the problem of causal imitation learning, where the goal is to construct a policy that replicates the outcomes of an expert.  The challenge is that the expert may e.g., make decisions based on unobserved variables.  Prior work (e.g., [36] as cited in this paper) has established graphical conditions under which imitation can still be achieved.  This paper shows that context-specific independence relations (or "CSIs"), if known, can expand the space of scenarios where imitation can be achieved, and give algorithms for performing imitation learning in this setting.  In addition, this paper gives an algorithm for potential identification when the graphical criterion fails (see Proposition 4.2, Alg 2, Theorem 4.3), although this algorithm is not necessarily complete, a limitation acknowledged in the conclusion


**Strengths:**

The contribution of this paper is quite clear, technically interesting, and novel.  Context-specific independence relations have been studied before in the causal inference literature (see e.g., citation [32]) in the context of causal identification, and so has the problem of causal imitation learning (see e.g., [36]). However, from my reading, this paper does more than simply put these concepts together (CSIs and causal imitation learning) in an obvious way. As it happens, this paper shows that incorporating CSIs renders the problem of imitation learning NP-hard

While there is a substantial amount of technical detail to parse, which perhaps makes the paper a bit difficult to skim, I found the paper to be reasonable clear in the technical presentation given a close read. Likewise, the authors clearly outline some of the limitations of their proposed approach, e.g., the fact that they provide a sound (but not necessarily complete) algorithm in Section 4.

The synthetic experiments are somewhat minimal, but I would not expect extensive experiments in a more theoretically-oriented paper like this one, and I found the setup of the synthetic experiments to be well-motivated, exploring the benefits of their approach over random graphs.


**Weaknesses:**

First, in order to be practically applicable, this method requires some knowledge of context-specific independence relations, where there is a hard independence between certain variables in certain contexts.

However, the motivating examples given in this paper for context-specific independence seem somewhat unrealistic.  The examples given in the paper are
* Lines 60-65: No impact of education on wages when unemployment is high
* Lines 66-70: In heavy traffic, no impact of speed limit on driving
* Lines 250-252: Company pricing is independent of demand during a recession

Of these, the second example seems most realistic.  In the others, complete independence between variables in those contexts seems unrealistic.

Second, I would not overstate the "straightforward" nature of solving for $\pi^*$ in Section 4 (see e.g., lines 257-259, "solving the aforementioned linear system of equations for $\pi^*$ is straightforward, for it boils down to a matrix inversion").  As mentioned in the footnote, this is only generally true in discrete settings, and even then may not be very practical with large numbers of variables, or variables with large cardinality.  Moreover, moving from the discrete to continuous setting introduces some substantial technical difficulties with e.g., ill-posed inverse problems.  This aspect is not the main focus of the paper, so I consider it a somewhat minor piece of feedback.

As an additional minor point, there is some lack of clarity in the experiments;  E.g., clarifying what is $\pi_{ALG}$ versus $\hat{\pi}_{ALG}$ in Table 1.  There are also some minor typos
* Line 25 "bypass IRL step" -> "bypass the IRL step"
* Line 28 "is for the most part result of" -> "is for the most part the result of"
* Line 75, missing space "For instance,[32]"


**Questions:**

The main weakness of this paper, in my view, is the plausibility of finding context-specific independences in real-world problems.

Are there other motivating examples, beyond those discussed already in the paper, that the authors would consider particularly compelling?

**Limitations:**

Yes

---

> ### Author Rebuttal · Authors · 2023-08-07
>
> We thank the reviewer for their thorough feedback. We acknowledge their positive feedback regarding the novelty and clarity of our work. We will now turn our attention to the points raised by the reviewer.
>
>
> ---
> >The main weakness of this paper, in my view, is the plausibility of finding context-specific independences in real-world problems. Are there other motivating examples, beyond those discussed already in the paper, that the authors would consider particularly compelling?
>
> In addition to the examples discussed in the paper, CISs have been utilized to analyze, for example, gene expression data [1], proteins [2], dynamics of pneumonia [3], parliament elections, prognosis of heart disease and occurrence of plants [4]. For a simple instance, consider an antibiotic that normally has a dose–response effect on the number of bacteria. However, due to a genetic mutation, the bacteria become resistant to the antibiotic, resulting in an independent relationship between the dose and the number of bacteria in the context of this mutation. As another example, In general, smoking has a causal effect on blood pressure. Nevertheless, if a person's ratio of beta and alpha lipoproteins exceeds a specific threshold, smoking is unlikely to have any significant impact on their blood pressure [5].
>
> Moreover, we envision that CSI relations can be particularly compelling in fields such as epidemiology, environmental sciences, public policy, and social sciences. For instance, in epidemiology, understanding the context-specific effects of certain risk factors on disease outcomes can be critical for designing targeted interventions and public health policies. In environmental sciences, the interactions between environmental variables and their effects on ecosystems may exhibit context-specific behavior, and CSIs can help unravel these complex relationships.
>
> Furthermore, in social sciences, studying the impact of social interventions or policies on different subgroups of a population may reveal context-specific causal patterns. For example, the effectiveness of a job training program may vary based on the participants' prior work experience, educational background, or age group.
>
> While the plausibility of finding CSIs in real-world scenarios may present challenges, we believe that their existence and relevance in multiple domains make them a compelling concept to explore and incorporate into causal modeling and inference approaches.
>
>
> [1] Y. Barash and N. Friedman. Context-specific Bayesian clustering for gene expression data.
> Journal of Computational Biology, 9(2):169–191, 2002.
>
> [2] B. Georgi, J. Schultz, and A. Schliep. Context-specific independence mixture modelling
> for protein families. In European Conference on Principles of Data Mining and Knowledge
> Discovery, pages 79–90. Springer, 2007.
>
> [3] S. Visscher, P. Lucas, I. Flesch, and K. Schurink. Using temporal context-specific independence
> information in the exploratory analysis of disease processes. In Conference on Artificial
> Intelligence in Medicine in Europe, pages 87–96. Springer, 2007.
>
> [4] H. Nyman, J. Pensar, T. Koski, and J. Corander. Stratified graphical models-context-specific
> independence in graphical models. Bayesian Analysis, 9(4):883–908, 2014.
>
> [5] Edwards, D. and Toma, H. (1985). A fast procedure for model search in multidimensional contingency tables. Biometrika.
>
> ---
> > As an additional minor point, there is some lack of clarity in the experiments; E.g., clarifying what is $\pi_{ALG}$
>  versus $\hat{\pi}_{ALG}$ in Table 1.
>
> We thank the reviewer for bringing this to our attention. We will provide the following clarification in the experiments section: $\pi$ refers to the policy that the algorithm would have learned with an infinite number of samples, while $\hat{\pi}$ is the policy it actually learns with the given finite sample size.
> By including this distinction in Table 1, we aimed to show that the sub-optimality of the naive algorithms is not due to the limited sample size but is inherent to the approach itself.
>
> We also thank the reviewer for other minor comments.

---

> > ### Comment · Reviewer_g52k · 2023-08-14
> >
> > Thank you for the thoughtful response!  If space permits, adding more concrete examples like those to the introduction would be helpful in my view.  I will maintain my generally positive score (Accept).

---

### Official Review · Reviewer_2EML · 2023-07-05

**Soundness:** 3 good
**Presentation:** 4 excellent
**Contribution:** 4 excellent
**Rating:** 7
**Confidence:** 3

**Summary:**

This paper explores the potential benefits of incorporating context-specific independence (CSI) information into causal imitation learning, where CSI relations are known. The authors prove that the decision problem for the feasibility of imitation in this setting is NP-hard, provide a necessary graphical criterion for imitation learning under CSI, and propose an algorithmic approach for causal imitation learning that takes both CSI relations and data into account.

**Strengths:**

**Clarity**

1. This paper is well-written and self-contained. It covers previous literature thoroughly.
2. The introduction section does an excellent job of motivating the research problem, and the problem statement is clear.

**Significance**

1. The research problem is interesting and significant in practice.

**Literature**

1. This paper covers related literature extensively.

**Soundness**

1. All results appear to be sound.

**Weaknesses:**

**Clarity in Assumptions**

1. I believe that the assumption used in Proposition 3.9 requires justification. Without an explanation, it is difficult to assess the generality of the assumption. It would be interesting if the simulation described in Section 5.1 provided the fraction of instances in which the assumption was satisfied.

**Lack of real-world dataset analysis.**

1. I believe the significance of the paper lies in its practical benefits of considering the Channel State Information (CSI), which is prevalent in the real world. Therefore, it would be beneficial to have a simulation scenario that incorporates a real-world dataset.

**Questions:**

- Q1. How strong is the assumption that "the context variables have parents only among the context variables"? It is difficult to discern the insights from which this assumption was generated without reasoning on it.
- Q2. Equation (1) is difficult to parse. What is V’ here? Could you simplify it or provide a verbal explanation?
- Q3. "The labels compatible with w" should be formally defined.
- Q4. Is "G_{w}" defined? Does it refer to the context-induced subgraph of G^{L} with respect to w?
- Q5. What is the computational cost for evaluating pi^{*} in Theorem 3.10? Isn't it still exponential in evaluating the equation?
- Q6. Is the result valid for continuous variables? It seems that the paper assumes discreteness throughout.

**Limitations:**

1. There is a lack of justification for the assumption, making it difficult to assess the clarity and significance of the statement.
2. I believe the significance of the paper lies in its practical benefits of considering the Channel State Information (CSI), which is prevalent in the real world. Therefore, it would be beneficial to have a simulation scenario that incorporates a real-world dataset.

---

> ### Author Rebuttal · Authors · 2023-08-07
>
> We thank the reviewer for their valuable input and appreciate the positive feedback regarding the significance of our work. We address the main points and questions raised by the reviewer below.
>
> ---
> **Regarding the assumption of Proposition 3.9**
> >I believe that the assumption used in Proposition 3.9 requires justification.
>
> >Q1. How strong is the assumption that "the context variables have parents only among the context variables"?
>
>
> The assumption used in Proposition 3.9 is based on the premise that the context variables are only influenced by other context variables and not by non-context variables. This assumption implies that non-context variables do not have a causal impact on the context variables.
> To illustrate this, consider the example discussed in Section 4, where a company aims to maximize revenue. It is reasonable to assume that factors such as pricing policy or demand rate, which are non-context variables, do not influence the context variable $C$ representing the macroeconomic variable of recession.
>
> However, it is important to note that if in some instances, certain context variables are influenced by factors other than $C$, the corresponding CSI relations can be disregarded without compromising the overall approach. In such cases, our approach remains applicable, although with a potential loss of imitability.
>
> In summary, while the assumption of having parents only among the context variables is ideal, our approach can still be adapted and remains valid in cases where this assumption does not strictly hold.
>
> ---
> > Q2. Equation (1) is difficult to parse. What is $V’$ here? Could you simplify it or provide a verbal explanation?
>
> $C(\mathcal{L})$ in equation (1) is the set of variables, at least one realization of which results in a context-specific independence (or removal of an edge, graphically speaking).
> In particular, $\mathbf{V}'$ is an arbitrary subset of nodes containing $V_i$ in Equation (1). The argument is that if there exists some arbitrary subset $\mathbf{V}'$ of nodes containing $V_i$ such that a realization $\ell$ of this subset ($\mathbf{V}'$) results in an independence (e.g., of some $V_j$ and $V_k$), then $V_i$ is considered as a context variable.
> We thank the reviewer for this feedback and we will simplify this equation and include further explanation in the revision.
>
> ---
> >Q3. "The labels compatible with w" should be formally defined.
>
> In our context, labels refer to realizations of a subset of variables. Formally, we say that a label $\ell$ is compatible with a realization $w$ if they are consistent, meaning they have the same value on the intersection of the variables to which they assign a value.
> We will add this explanation in the revision.
>
> ---
> > Q4. Is "G_{w}" defined? Does it refer to the context-induced subgraph of G^{L} with respect to w?
>
> Yes, indeed. We thank the reviewer for pointing this out, and we will clarify that this symbol refers to the context-induced subgraph of $\mathcal{G}^\mathcal{L}$ w.r.t. $\mathbf{w}$, as defined in definition 3.5.
>
> ---
> >Q5. What is the computational cost for evaluating pi^{*} in Theorem 3.10? Isn't it still exponential in evaluating the equation?
>
> Algorithm 1 runs linearly many loops in terms of the number of contexts. The loop itself requires testing a d-separation which is quadratic-time in the worst case, in the number of variables. The number of possible contexts, however, can still be exponential in the number of variables. This is inevitable considering the hardness result of Theorem 3.8.
>
> ---
> >Q6. Is the result valid for continuous variables? It seems that the paper assumes discreteness throughout.
>
> The discreteness assumption can indeed be relaxed under certain considerations.
> In particular, Algorithm 1 requires assessing the d-separation of line 5 in the context-induced subgraphs $\mathcal{G}_c$.
> Even when the context variables $C$ are continuous, the domain of these variables can be partitioned into at most $2^m$ equivalence classes in terms of their corresponding context-induced subgraph, where $m$ denotes the number of labeled edges.
> This holds since the number of context-induced subgraphs cannot exceed $2^m$.
> It is noteworthy, however, that solving the equation referred to in line 10 of Algorithm 2 for continuous variables may bring additional computational challenges. We will elaborate on this point in the revision.

---

> > ### Comment · Reviewer_2EML · 2023-08-13
> > **Response to the rebuttal**
> >
> > The authors' rebuttal adresses my questions and concernts. I will maintain the positive assessment.

---

### Official Review · Reviewer_Fd5v · 2023-07-06

**Soundness:** 4 excellent
**Presentation:** 3 good
**Contribution:** 2 fair
**Rating:** 7
**Confidence:** 4

**Summary:**

This paper extends studies on causal imitation learning to settings in which additional information can be provided in the form of context-specific independences (CSIs). Causal imitation learning seeks to maximize some unobserved reward $Y$ by finding a policy $\pi^*$ from the space of policies $\Pi$ such that the reward distribution under that policy, $P(Y \mid do(\pi))$ matches that of the expert policy, $P(Y)$, thereby mimicking the expert. However, mimicking $P(Y)$ is not always possible in settings with unobserved confounding, so additional knowledge in the form of causal constraints is necessary to decide whether $P(Y)$ is imitable. In prior works where such constraints are assumed in the form of a causal diagram $\mathcal{G}$, it has been proven that $P(Y)$ is imitable if and only if there exists a $\pi$-backdoor admissible set $\mathbf{Z}$ w.r.t. $\langle \mathcal{G}, \Pi \rangle$. However, completeness no longer holds when CSIs are provided, in which further independence information about the distributions may be provided, conditioned on specific settings of the variables. The paper first proves that deciding imitability in this more general setting is NP-hard (Thm. 3.8). Given this limitation, they provide a sound algorithm that checks imitability within each context-induced subgraph, which they prove is also complete under a further assumption (Alg. 1). In settings where this assumption does not hold, they show that imitability can still be achieved if a policy on a set of surrogate variables is identifiable under a specific setting of CSIs. This provides a more general algorithm (Alg. 2), evaluated experimentally.

**Strengths:**

To be fully transparent, I have reviewed this paper in the past, so the following list of strengths and weaknesses reflect my existing impressions of this paper, which I have updated following another read of the latest manuscript. However, my points are largely the same, since I do not see many notable changes in the latest version. Please correct me if I am wrong.

**Strengths:**
1. Problem is well-motivated. LDAGs arise in practice and provide more information than standard causal diagrams, which should be leveraged to allow more imitable cases. This would add more positive results to the causal imitation learning literature.


2. Assumptions are clearly stated. LDAGs are well defined, and the paper does a good job of explaining specifically how the additional constraints are incorporated. The authors are very transparent with the limitations of their approach.


3. The solution is nontrivial and interesting. The paper clearly shows non-imitable cases that are rendered imitable when accounting for CSIs. The obvious solution of checking for backdoor sets in each context-induced subgraph is only complete under a strict assumption.

**Weaknesses:**

I reiterate that these points are similar to the points I made in my previous read of the paper. I have an overall positive opinion of the paper, and my goal is to help improve the paper, so I hope the authors will take this feedback into consideration.

**Weaknesses:**
1. There is nothing done to address the NP-hardness claimed by Thm. 3.8. Alg. 1 and 2 still take exponential time in the worst case. It may be insightful to provide some alternative settings in which assumptions are strict enough that polynomial time solutions can be developed. Alternatively, algorithms that sacrifice completeness for speed could be provided. While I understand that this may be out of the scope of the paper, it is otherwise not clear to me why Thm. 3.8 is relevant to the paper in the first place.


2. Unfortunately, even Alg. 2 is not complete in the general case. Completeness is not a requirement for it to serve as a real contribution, but it would help to have some insights in this paper on why Alg. 2 is incomplete (e.g. some counterexamples) and some ideas on what could be done to move towards completeness.


3. The motivation for Alg. 2 could be improved in terms of clarity. Notably, it could be emphasized why Alg. 1 fails in a more general setting. Eq. 3 could be explained better as well to motivate the idea of context-specific surrogates (I did not really understand Eq. 3 until I derived it myself by hand).


4. The experiments illustrate the point as intended, but the tested scenarios are very limited in scope. The first experiment only studies a specific family of graphs, and the second experiment is performed on one specific SCM. Neither of these choices are justified. While a more extensive empirical study would boost the strength of this work, it would help immensely just to be transparent about the data generating process to ensure that there was no cherry picking. For example, for Sec. 5.1, why were those choices of delta and probability of latent variables chosen? And for Sec. 5.2, why were the parameters for that specific SCM chosen (as described in Appendix C)?


5. Many of the ideas in this paper (including the interesting point about surrogates) incrementally improve existing ideas from Zhang et al. (2020). This reduces some of the novelty.


6. This is a minor point and did not affect my judgment of the score, but on line 261, the authors describe the imitability problem under CSIs using the inputs of $\langle \mathcal{G}^{\mathcal{L}}, \Pi, P(\mathbf{O}) \rangle$ as opposed to $\langle \mathcal{G}^{\mathcal{L}}, \Pi \rangle$. I understand that this is due to the CSI setting requiring additional information in the form of constraints over $P(\mathbf{O})$, but I think this could be better framed. Both the original imitability problem and the new version with CSIs uses $P(\mathbf{O})$, since it must be clear that only observational data is available, as opposed to additional interventional data such as $P(\mathbf{O} \mid do(\mathbf{x}))$, collected from experimentation. In addition to this however, the CSI setting should include a set of independences.


Overall, my impression is that this paper is worth publishing, since everything is well-defined, the proofs make sense, the assumptions are clear, and the claims are sufficiently backed. I believe that a score of 6 is appropriate given the level of contribution of the paper, which is solid but somewhat limited.

**Questions:**

No questions, but would be interested in hearing author responses in case I missed something.

**Limitations:**

Limitations are clearly stated.

---

> ### Author Rebuttal · Authors · 2023-08-08
>
> We thank the reviewer for their valuable input.
>
> **Regarding NP-hardness and polynomial-time solutions**
>
> We agree that providing settings where a polynomial-time solution exists is insightful, and we thank the reviewer for this suggestion. We have come up with certain restrictions that allow for polynomial-time solutions. One such case is bounded-degree graphs. We will discuss these settings along with their corresponding polynomial-time solutions to provide further insights to the reader.
>
> ---
> **Regarding Algorithm 2**
>
> We appreciate the reviewer's understanding that completeness is a challenging requirement for the proposed Algorithm 2. While achieving completeness for imitation in this setting poses significant difficulties, we acknowledge the importance of providing insights and counterexamples to aid the readers in understanding the limitations and potential improvements of our work.
> In the revised version of the paper, we will include a discussion of counterexamples and scenarios where Algorithm 2 may not produce the optimal solution.
>
> As a concrete example, take for instance the instrumental variable graph ($C\gets X\gets U\to Y$, $X\to Y$), where $U$ is an unobserved confounder, $Y$ is a binary unobservable reward, and $C$ and $X$ are observed binary random variables. Also suppose that the edge $U\to X$ has the label $C=0$. This is similar to our Figure 3 (a) in the text, without the surrogate $S$.
> It is evident that if the set of equations
>
> $$\left(P(y\vert do(\pi))=\right)P(y\vert X=0, C=0)\pi(X=0)+P(y\vert X=1, C=0)\pi(X=1) = P(y\vert X=0)P(X=0)+P(y\vert X=1)P(X=1) \left(=P(y)\right)$$
> for both values of $y=0$ and $y=1$ is a set of linear equations with two equations in two parameters, namely $\pi(X=0)$ and $\pi(X=1)$, which always has a solution.
> This indeed indicates that this instance is imitable, i.e., there always exists at least one imitation policy. However, Algorithm 2 would not be able to find any such policy.
>
> ---
> **Regarding the experiments**
>
> Indeed our experiments were primarily aimed at illustrating the potential benefits of taking context-specific information (CSIs) into account in the imitation learning process, particularly in terms of 'achievability' of imitation learning. We acknowledge that the current experiments are limited to synthetic datasets, and we agree that testing the proposed algorithms on more extensive datasets is essential to validate their practical applicability. As we move forward, we plan to address this limitation. Developing more efficient algorithms for handling CSIs as well as conducting extensive experiments on real-world datasets to evaluate the performance and efficiency of our approach in practical scenarios will be a part of our future work to enhance the overall usability and scalability of the proposed methods.
>
> We also thank the reviewer for other minor points and comments.

---

> > ### Comment · Reviewer_Fd5v · 2023-08-14
> > **RE: Rebuttal by Authors**
> >
> > I have read the rebuttal, and I thank the authors for addressing my concerns. I will raise my score to a 7 under the assumption that the authors will add the promised revisions to the paper.

---

### Decision · Program_Chairs · 2023-09-21

**Decision:**

Accept (poster)

**Comment:**

Unanimous reviewer opinion after discussions with upward score revisions.

Authors deal with imitation learning with context dependent independencies based side information. Authors show NP-hardness result regarding decision problem of feasibility of imitation learning which is interesting. Authors show necessary and sufficient graphical criterion given the side information for feasibility of imitation learning.

Authors have promised a few revisions in the new camera ready. I strongly suggest authors to pay attention to these.